# Integrating Single-Cell RNA-Seq and Bulk RNA-Seq Data to Explore the Key Role of Fatty Acid Metabolism in Breast Cancer

**DOI:** 10.3390/ijms241713209

**Published:** 2023-08-25

**Authors:** Yongxing Chen, Wei Wu, Chenxin Jin, Jiaxue Cui, Yizhuo Diao, Ruiqi Wang, Rongxuan Xu, Zhihan Yao, Xiaofeng Li

**Affiliations:** Department of Epidemiology and Health Statistics, Dalian Medical University, Dalian 116044, China

**Keywords:** fatty acid metabolism, immunotherapy, breast cancer, tumor microenvironment, single-cell sequencing

## Abstract

Cancer immune escape is associated with the metabolic reprogramming of the various infiltrating cells in the tumor microenvironment (TME), and combining metabolic targets with immunotherapy shows great promise for improving clinical outcomes. Among all metabolic processes, lipid metabolism, especially fatty acid metabolism (FAM), plays a major role in cancer cell survival, migration, and proliferation. However, the mechanisms and functions of FAM in the tumor immune microenvironment remain poorly understood. We screened 309 fatty acid metabolism-related genes (FMGs) for differential expression, identifying 121 differentially expressed genes. Univariate Cox regression models in The Cancer Genome Atlas (TCGA) database were then utilized to identify the 15 FMGs associated with overall survival. We systematically evaluated the correlation between FMGs’ modification patterns and the TME, prognosis, and immunotherapy. The FMGsScore was constructed to quantify the FMG modification patterns using principal component analysis. Three clusters based on FMGs were demonstrated in breast cancer, with three patterns of distinct immune cell infiltration and biological behavior. An FMGsScore signature was constructed to reveal that patients with a low FMGsScore had higher immune checkpoint expression, higher immune checkpoint inhibitor (ICI) scores, increased immune microenvironment infiltration, better survival advantage, and were more sensitive to immunotherapy than those with a high FMGsScore. Finally, the expression and function of the signature key gene *NDUFAB1* were examined by in vitro experiments. This study significantly demonstrates the substantial impact of FMGs on the immune microenvironment of breast cancer, and that FMGsScores can be used to guide the prediction of immunotherapy efficacy in breast cancer patients. In vitro experiments, knockdown of the *NDUFAB1* gene resulted in reduced proliferation and migration of MCF-7 and MDA-MB-231 cell lines.

## 1. Introduction

In recent years, the incidence of breast cancer (BC) has surpassed that of lung cancer as the most prevalent cancer worldwide [1], with an estimated 2.3 million new cases per year (11.7% of new cancer cases) and its incidence is increasing [2]. The development of diagnostic tools and imaging techniques has significantly improved the early detection rate of breast cancer and reduced the mortality rate. However, breast cancer is a highly heterogeneous cancer, making the treatment of patients challenging and leading to poor prognosis. Therefore, there is an urgent need for new biomarkers to guide breast cancer treatment. Breast cancer progression involves significant metabolic reprogramming to support tumor cell growth. Fatty acid metabolic reprogramming is particularly critical for cancer cells as they rely on fatty acid metabolism for energy production, signaling, and cell membrane formation [3] (p. 36). The proportion of saturated, monounsaturated, and polyunsaturated fatty acids in membranes is now recognized as crucial for promoting cell survival and preventing lipotoxicity and ferroptosis [4]. Moreover, fatty acid metabolism has been shown to play a significant role in the differentiation and migration of tumor-associated immune cells [5]. TME refers to the microenvironment surrounding tumor cells, consisting of endothelial cells, immune cells, stromal fibroblasts, various signaling molecules, cytokines, and the extracellular matrix [6]. Numerous studies have linked the TME to the development of breast cancer [7], including the combination of suppressive immune cells, soluble factors, and altered extracellular mesenchyme, leading to immune escape and promoting the progression and metastasis of breast cancer [8]. The major modalities of tumor immunotherapy currently include immune checkpoint inhibitor therapy (ICB), adoptive cell transfer, and tumor vaccination strategies [9], and ICB, in particular, has become the latest treatment for various cancers [10]. However, the efficacy of immunotherapy in breast cancer is lower than in other cancers [11,12]. Studies have demonstrated that metabolic reprogramming in the TME contributes to immune escape [13], especially fatty acid metabolic reprogramming, which affects the differentiation of relevant immune cells and tumor cell migration in the tumor microenvironment [14]. Identifying metabolic targets with prognostic impact on breast cancer and combining them with immunotherapy may enhance immune efficacy in breast cancer patients. We identified 15 fatty acid metabolic genes associated with breast cancer prognosis in this study and comprehensively evaluated the impact of fatty acid-related gene expression on mutation, prognosis, and immune response pathways in breast cancer. Unsupervised clustering was used to identify three different fatty acid patterns of breast cancer and assess the differences in prognosis and immune infiltration pathways among them. The three FMG patterns were closely associated with the three immune types. In addition, we classified patients into three different genomic subgroups based on differentially expressed genes (DEGs) related to the FMG patterns. Furthermore, we established a FMGsScore that effectively predicted prognosis and immunotherapy response in breast cancer patients.

## 2. Results

### 2.1. Identification of Prognostic Fatty Acid Metabolism-Related Genes and Genetic Variation in Breast Cancer

The flow chart of this study is shown in Figure 1. Differential analysis of 309 FMGs in TCGA, using thresholds |log2FC| > 0.585 and FDR < 0.05, detected 121 differentially expressed genes (DEGs) by comparing 1082 breast cancer samples with 113 normal breast samples. In the DEGs, there were 50 significantly enhanced FMGs in BC patients, while 71 FMGs in BC samples were largely attenuated (Figure 2A,B). To assess the potential prognostic value of each FMG, univariate Cox risk regression analysis was performed on the DEGs to identify FMGs associated with overall survival (OS). A total of 15 significant OS-related FMGs were selected for further analysis (Figure 2C). The reciprocal network between these 15 FMGs is presented (Figure 2D). The FMG modulator network provides an overview of mutual effects, modulator associations, and prognoses for patients with breast cancer. Somatic mutations and copy number variants of the 15 FMGs in the TCGA cohort were analyzed. Among the 986 samples, only 21 (2.13%) exhibited mutations in FMGs, specifically *CYFIP1*, *LARP1*, *EIF4G3*, and *AGO2*. The overall mutation frequency was low, ≤1% (Appendix A). Location of CNV alterations in 15 fatty acid-related genes altered on 23 chromosomes (Appendix A). In comparison to other FMGs, *ACSL1*, *ACSL5,* and *ALOX15B* exhibited a higher frequency of CNV deletion, whereas *UBE2L6*, *HSPH1,* and *PSME1* showed a higher frequency of CNV amplification (Appendix A).

### 2.2. Immune Infiltration and Biological Functions Associated with FMG Modification Patterns

In our analysis, we used two breast cancer datasets (TCGA-BRCA, GSE42568), OS, and clinical data. To identify patients with distinct FMG modification patterns, we quantified the expression levels of the 15 FMG modulators using the R software (4.3.0) of Consensus Cluster Plus. Three different modification patterns were identified via unsupervised clustering: 477 patients with pattern A, 418 patients with pattern B, and 261 patients with pattern C (Figure 3A,B, Appendix A). We further validated the clustering effectiveness through the METABRIC dataset (Appendix A). The survival analysis revealed no significant difference in survival among the three patterns (Figure 3C). Therefore, we aimed to explore the biological behaviors of different FMG patterns to understand the underlying reasons. To explore the biological behavior of different FMG patterns, we performed an enrichment analysis using GSVAs (Figure 3D–F). FMGsCluster-A showed significant enrichment in the matrix and oncogenic activation pathways, including ECM receptor interactions, TGF β signaling pathway, cell adhesion, JAK-STAT signaling pathway, activation of the chemokine signaling pathway, cytokine-cytokine receptor interactions, T-cell receptor signaling pathway, and Toll-like receptor signaling pathway. Additionally, activation-related pathways were also enriched (Figure 3D). FMGsCluster-B did not show any association with the enrichment of immune-related activation pathways (Figure 3E). In contrast, FMGsCluster-C exhibited activation of the chemokine signaling pathway, cytokine-cytokine receptor interaction, T-cell receptor signaling pathway, and B-cell receptor signaling pathway (Figure 3F). Subsequent analysis of FMGsCluster cell infiltration in the tumor microenvironment (TME) of breast cancer patients using ssGSEA revealed that FMGsCluster-A exhibited higher natural immune cell infiltration compared to FMGsCluster-B and FMGsCluster-C. Specifically, FMGsCluster-A showed the highest level of immune cell infiltration (Appendix A). FMGsCluster-A showed enrichment in natural immune cell infiltration, comprising B cells, CD4+ T-cells, CD8+ T-cells, natural killer cells, macrophages, eosinophils, mast cells, MDSC, and plasmacytoid dendritic cells (Appendix A). However, patients with FMGsCluster-A did not show a corresponding survival advantage (Figure 3C). The immune exclusion phenotype is a type of immune phenotype characterized by the infiltration of intrinsic immune cells in the tumor microenvironment (TME). However, these immune cells do not penetrate the core of the tumor cell nests but instead remain in the stroma surrounding the tumor cell nests. Consequently, due to the activation of stromal pathways, these immune cells may not effectively exert anti-tumor effects on the tumor cells [15,16]. Therefore, the formation of the immune exclusion phenotype is considered to be associated with stromal activation-induced immune suppression in the TME. The activation of the associated stromal pathway is believed to result in T-cell suppression. The GSAV results showed that FMGsCluster-A was also significantly enriched in the stromal-activated pathway, leading to speculation that stromal activation in FMGsCluster-A might suppress the tumor-killing effect of its immune cells. The subsequent analysis also showed that FMGsCluster-A had significantly higher activation in cell adhesion, transforming growth factor β (TGFb), and angiogenic pathways than FMGsCluster-B and FMGsCluster-C (Appendix A). We classified FMGsCluster-A as an immune rejection phenotype, which is characterized by innate immune cell infiltration and stromal activation; FMGsCluster-B as an immune desert phenotype, characterized by immunosuppression; and FMGsCluster-C as an immune inflammatory phenotype, characterized by adaptive immune cell infiltration and immune activation. The Molecular Taxonomy of Breast Cancer International Consortium (METABRIC) database, consisting of 1904 breast cancer samples, was utilized to validate the applicability of 15 Functional Molecular Groups (FMGs) for classification purposes (Appendix A). The results indicate that the 15 Functional Molecular Groups (FMGs) can categorize the METABRIC data into four distinct classes.

### 2.3. Construction of Gene Signatures Based on Differential Genes of FMGsCluster

To explore the biological features of the potential regulation among the three FMGsClusters, we identified 898 DEGs associated with the three FMGsClusters using the “limma” package (Figure 4A). GO enrichment analysis showed that 898 DEGs were significantly associated with immunity in terms of cellular components (CC), molecular functions (MF), and biological processes (BP) (Figure 4B). Additionally, the KEGG enrichment analysis indicated that these genes were significantly enriched in immune pathways, including cytokine-cytokine receptor interaction, natural killer cell-mediated cytotoxicity, and T-cell receptor signaling pathway, among others (Figure 4C). The above results suggest the critical role of fatty acid metabolism genes in immune cell function within the tumor microenvironment. To comprehend and validate their regulatory pathways, the 898 genes related to fatty acid metabolism were subjected to unsupervised clustering analysis, enabling the classification of patients into distinct genomic subtypes. In line with the clustering grouping of FMG modification patterns, the unsupervised clustering algorithm also revealed three distinct genomic phenotypes, which we designated as FMG gene clusters A, B, and C, respectively. Survival analysis indicated that gene cluster A exhibited a survival advantage over gene cluster B and gene cluster C (Figure 4D). The heat map comparing clinical information among the three gene clusters (gene cluster A, gene cluster B, and gene cluster C) indicates that gene cluster A and gene cluster B exhibit distinct clinical characteristics (Figure 4E). The expression of the 15 FMGs in the three gene clusters was significantly different (Appendix A). These results collectively indicate the presence of three distinct patterns of fatty acid metabolic modifications in breast cancer patients. To further explore the role of FMGs in breast cancer, differentially expressed genes between FMG expression patterns were identified using the “limma” R package. Significant genes were filtered with *p* < 0.001, resulting in 898 common differentially expressed genes across the three FMG clusters. Univariate Cox regression analysis identified 210 prognosis-related genes. Principal component analysis (PCA) was utilized to construct the FMGsScore. The survival differences between the high and low FMGsScore groups were analyzed using K–M curves, and the results showed that the low-scoring group had a better prognosis (Figure 5A). Furthermore, it is evident that both FMGsCluster-B and gene cluster B exhibited higher FMGsScore scores (Figure 5B,C), and both FMGsCluster-B (Figure 3C) and gene cluster B had a worse prognosis (Figure 4D). The Spearman correlation analysis between FMGsScore and immune cell infiltration revealed a negative correlation, indicating that FMGsScore was negatively associated with most of the immune cell infiltration. Furthermore, as the score decreased, the proportion of immune cell infiltration in patients increased (Figure 5D). Additionally, the allogram (Figure 5E) showed that the majority of the genes in FMGsCluster-A were grouped into geneClusters-A and C. Almost half of the genes in FMGsCluster-B were assigned to geneCluster-B, and a small portion of the remaining half were assigned to geneCluster-C. On the other hand, most of the genes in FMGsCluster-C were grouped into geneCluster-C, and a small portion of the remaining half were assigned to geneCluster-B. Both geneClusters-B and C were grouped into the high FMGsScore group. The low FMGsScore group had a higher survival rate, whereas the high FMGsScore group had a lower survival rate.

### 2.4. Clinical and Tumor Somatic Cell Mutation Characteristics between Patient High and Low FMGsScore Groups

The low subgroup exhibited a higher proportion of surviving patients, while the patients who died had a higher FMGsScore (Figure 6A). The age distribution of patients was comparable between the low and high FMGsScore subgroups. However, the high age group exhibited a higher FMGsScore as depicted in Figure 6B. The proportion of patients with high FMGsScore was higher in stage 1–2 and N1-3 in patient stage grading and N grading. In patient subgroups, a high FMGsScore was significantly associated with poor prognosis in patients with stage 1–2, T1-2, and N0 (Figure 6C–E). Next, we assessed the FMGsScore and tumor mutational burden and their association with patient prognosis. The R package “maftools” was used to create a panorama of mutations in somatic cells between the high and low FMGsScore groups. The mutation rates of somatic cells were comparable between the high and low groups, with rates of 82.5% and 90.7%, respectively (Figure 7A,B). Secondly, the difference in the tumor mutation burden (TMB) between high and low groups was not statistically significant (Figure 7C), and the correlation analysis between FMGsScore and TMB also showed no statistically significant negative correlation (Figure 7D), but the results of survival analysis showed that the higher the TMB, the worse the prognosis (Figure 7E). Among them, patients with both high TMB and a high FMGsScore had the worst prognosis (Figure 7F).

### 2.5. Effect of FMGsScore on Immunotherapy

ICB therapy has become the main modality of immunotherapy for many cancers. Therefore, we analyzed the differences in the expression levels of some immune checkpoint genes (PD1, PD-L1, CTLA4, LAG3, HAVCR2, and IDO1) between the high and low FMGsScore groups. We found that the expression of immune checkpoint genes was higher in all of the low FMGsScore group than in the high FMGsScore group (Figure 8A). ICB therapy with PD1 and CTLA4 blockade is an effective treatment for certain cancers. To explore the relationship between FMGsScore and immunotherapy, we downloaded immunotherapy data from the TCIA database for BC patients in all of the ctla4_neg_pd1neg, CTLA4_neg_PD1_pos, CTLA4_pos_PD1_neg, and ctla4_pos_pd1_pos groups. Patients with lower FMGsScore had a higher IPS, indicating that these patients had a better response to immunotherapy (Figure 8B). The comparison of immune cell infiltration abundance between the high and low FMGsScore groups revealed that the low FMGsScore group exhibited higher immune cell infiltration abundance than the high FMGsScore group (Appendix A). GSVA enrichment analysis revealed pathway differences between the high FMGsScore group and the low FMGsScore group. Compared with the high group, the low FMGsScore group had higher enrichment in tumor- and immune-related pathways, such as primary immunodeficiency, B-cell receptor signaling pathway, chemokine signaling pathway, cytokine receptor interactions, natural killer cell-mediated cytotoxicity, JAK/STAT signaling pathway, T-cell receptor signaling pathway, etc. (Appendix A). Hence, tumor patients with a low FMGsScore exhibited elevated fatty acid metabolism, extensive immune cell infiltration, increased expression of immune checkpoints, favorable prognosis, and heightened response to immunotherapy. The validity of the FMGsScore was confirmed using the GSE20685 dataset. The expression levels of six immune checkpoint suppressor genes were significantly higher in the low FMGsScore group compared to the high FMGsScore group, similar to the training group (Appendix A). Moreover, patients in the low FMGsScore group exhibited a survival advantage over those in the high FMGsScore group, as evident from the survival curves with fewer deceased patients. This further corroborated the accuracy of FMGsScore in the training group (Appendix A). In order to further validate the predictive capability of FMGsScore for immunotherapy response in the immunotherapy cohort treated with ICI, we expanded the study to include the metastatic melanoma immunotherapy cohort and the IMvigor210 cohort. This expansion allowed us to explore the prognostic significance of FMGsScore and its predictive ability for immunotherapy response. Patients with a low FMGsScore in both the metastatic melanoma cohort and the IMvigor210 cohort exhibited significantly higher overall survival compared to those with a high FMGsScore. Additionally, a higher proportion of patients with a low FMGsScore responded to immunotherapy (Appendix A).

In addition to this, lipid metabolism is also altered in different subtypes of breast cancer. Compared to other subtypes, Luminal A tumors typically exhibit lower lipogenesis and higher fatty acid oxidation rates. On the other hand, HER2-enriched and triple-negative breast cancers often show increased lipogenesis, which is associated with their aggressive behavior [17]. Therefore, we evaluated whether FMGsScore is influenced by specific subtypes (Appendix A). We found that although fatty acid metabolism varies among different subtypes, FMGsScore still retains the ability to predict the prognosis and immunotherapy response of patients across all breast cancer subtypes.

### 2.6. Single-Cell Profiling Data Reveal the Relationship between 15 Fatty Acid Metabolism Genes and Tumor Immunity

The single-cell sequencing data annotation identified a total of eight cell types. t-SNE plots (Figure 9A) depict cell subtypes in breast cancer samples, and a heat map of marker genes represents part of the eight annotated cell types (Figure 9B). Based on the detection of significant chromosomal copy number variations (CNV) in comparison to a reference dataset of normal epithelial cells, we successfully identified a cumulative count of 11,041 malignant epithelial cells within breast cancer tumors (details provided in Methods; depicted in Figure 9C,D). Stacked plots (Figure 9E) reveal a higher percentage of T-cell and myeloid infiltration in patients with triple-negative breast cancer compared to other breast cancer types, and a higher percentage of T cells in HR+ patients than in ER+ patients. The expression of 15 genes related to fatty acid metabolism was scored between cells using the function “Addmodulescore”, available in the R package “Seurat”. The cells were divided into two groups based on the average score. Furthermore, t-SNE plots illustrate the fatty acid metabolism of cell subtypes in breast cancer samples, and the t-SNE plot displays the fatty acid metabolism gene scores of the cell subtypes in the breast cancer samples (Figure 9F). Triple-negative breast cancer patients’ scores in the six samples were higher than those of other patients and normal samples (Figure 9G). Additionally, the scores of T-cells and myeloid cells were higher than those of other cells (Figure 9H). To explore intercellular interactions in the high- and low-score groups, we utilized the R package “CellChat” to infer intercellular interactions based on ligand-receptor signaling. The Circos plot illustrates the number of intercellular interactions between the high-score group and the low-score group (Figure 10A,B). Notably, the high-score group exhibited more interactions between cells compared to the low-score group, with stronger interaction strengths (Figure 10C). Furthermore, interactions between epithelial and endothelial cells were enhanced in the low-score group relative to the high-score group, whereas interactions between immune cells (T-cells, mast cells, and myeloid cells) were less pronounced than in the high-score group (Figure 10D). We compared cell-to-cell interaction pathways between the low-score group and the high-score group, and observed enhanced *MHC-I* and *MHC-II* pathway signaling in the high-score samples. This suggests the activation of antigen-presenting functions of CD8+ T-cells and CD4+ T-cells in the high-score samples, as well as the activation of *CXCL*-related pathways, such as *CXCL9-CXCR3*, *CXCL10-CXCR*, and *CXCL13-CXCR* (Figure 10E). The *CXCL9*, *-10*, *-11/CXCR3* axis primarily regulates immune cell migration, differentiation, and activation. This axis is known to recruit immune cells, including cytotoxic lymphocytes (CTL), natural killer (NK) cells, NKT cells, and macrophages [14]. Additionally, interactions related to interferon-mediated signaling pathways and antigen processing and presentation (*IFNG-IFNG R1*, *IFNG-IFNGR2*, *HLA-DRB1-CD4*, *HLA-DRA-CD4*, *HL A-DRPB1-CD4*, and *HLA-DRA1-CD4*) were also elevated in the high-score group (Figure 10F). This suggests that immune pathways are more active in patients with high expression of 15 fatty acid-related genes.

### 2.7. Critical Role of NDUFAB1 Gene in Migration and Proliferation in Breast Cancer Cells

The TCGA data confirmed a significant upregulation of the *NDUFAB1* gene in tumor samples (Figure 11A). Furthermore, patients with high *NDUFAB1* expression exhibited a poorer prognosis (Figure 11B). Among the 15 FMGs employed to formulate the FMGsScore, *NDUFAB1* exerts the most substantial prognostic influence and remains relatively unexplored in its role within cancer. Given its positive correlation with FMGsScore (Figure 11C), we opt to perform in vitro experiments to authenticate its involvement in breast cancer. Through in vitro experiments, we demonstrated that *NDUFAB1* promotes proliferation and migration of breast cancer cells. Western blot analysis confirmed the upregulation of the *NDUFAB1*-encoded protein in breast cancer, particularly in the MCF-7 cell line (Figure 11C). Additionally, Figure 11D and 11E showed a significant decrease in *NDUFAB1*-associated protein expression in transfected MCF-7 and MDA-MB-231 cells (Figure 11D,E).The qRT-PCR validated the upregulation of NDUFAB1 in both breast cancer cell lines (Figure 11F,G). Moreover, Figure 11H and 11I demonstrated a significant reduction in *NDUFAB1* expression after transfection in MCF-7 and MDA-MB-231 cells (Figure 11H,I). Moreover, subsequent analysis revealed a significant positive correlation between *NDUFAB1* and fatty acid elongation (Figure 12A). Long-chain fatty acids have been strongly associated with cancer development [18,19]. CCK8 analysis indicated that knockdown of *NDUFAB1* significantly reduced cell viability in both MCF-7 and MDA-MB-231 cell lines, suggesting that *NDUFAB1* may promote proliferation (Figure 12B,C). Furthermore, the wound healing assays in Figure 12D–G revealed that cells migrated more slowly after *NDUFAB1* knockdown compared to the NC and siNC groups, implying that *NDUFAB1* may promote migration of breast cancer cells.

## 3. Discussion

Breast cancer is the most prevalent cancer in women worldwide. While immunotherapy has shown efficacy for many cancer patients, breast cancer patients rarely benefit from it. Therefore, there is a need to identify new genes that can predict better responses to immunotherapy. At this stage, constructing models to screen for molecular targets with prognostic and therapeutic effects on breast cancer using various approaches has become a major focus of research. The breast is a fat-rich tissue with a relatively active fatty acid metabolism. However, there are limited studies on the comprehensive regulatory mechanisms of fatty acid metabolism genes in breast cancer. Tang et al. conducted a study screening fatty acid metabolism genes associated with overall survival (OS) in breast cancer patients and constructed a prognostic model using lasso regression [20]. Although this is similar to our study, our research specifically focuses on the relationship between fatty acid metabolism and the tumor microenvironment. Our model was constructed through unsupervised cluster analysis of typing, screening for differential genes, and scoring the model by PCA based on differential genes associated with prognosis. The advantage of our approach is that it incorporates more differential genes associated with fatty acid metabolism, rather than studying fatty acid metabolism genes in isolation. In this study, we screened for differential genes in breast cancer patients and normal samples based on 309 genes related to fatty acid metabolism. We extracted 15 genes (*ALOX15B*, *CEL*, *UBE2L6*, *ABCD1*, *HSPH1*, *PTGDS*, *NDUFAB1*, *ALDH3A1*, *ADH6*, *ACSL1*, *PSME1*, *NUDT19*, *ACSL5*, *RDH16*, and *IL4I1*) with prognostic impact on breast cancer patients, and these genes have been studied in the relevant literature. As an illustration, arachidonate 15-lipoxygenase type B (ALOX15B) falls within the lipoxygenase family of structurally akin nonheme iron dioxygenases. It participates in generating fatty acid hydroperoxides, particularly within tumor-associated macrophages (TAMs), potentially exerting a noteworthy influence on the synthesis of bioactive lipid mediators within the tumor microenvironment (TME). Blocking *ALOX15* reduces the infiltration of MDSCs or monocytes, thereby decreasing the number of tumor-associated macrophages (TAMs) [21]. Additionally, as a gene related to ferroptosis, it may act as an independent prognostic factor for breast cancer [22]. ATP binding cassette subfamily D member 1 (*ABCD1*) encodes a protein that belongs to the ATP-binding cassette transporter protein superfamily (ABC transporters). This protein is part of the ALD subfamily and is involved in the peroxisomal import of fatty acids and/or fatty acyl-coenzyme A into the organelle [23]. The observed downregulation of *ABCDs* in certain tumor types may lead to lipid accumulation, potentially promoting tumor growth and progression similar to X-linked adrenoleukodystrophy (X-ALD), through oxidative stress and inflammatory stimuli leading to peroxisome dysfunction [24].

In this study, we first investigated genetic variants in fatty acid metabolism-related genes, which are dysregulated in breast cancer patients and associated with prognosis, even though the frequency of mutations in 15 fatty acid metabolism-related genes is low. Then, we performed unsupervised cluster analysis to categorize BC patients into three groups, yielding three distinct fatty acid gene patterns (FMGsCluster-A, FMGsCluster-B, and FMGsCluster-C). Due to the complexity of interactions among fatty acid metabolism-related genes, we identified 898 differentially expressed genes (DEGs) based on these patterns (Figure 4A). GO enrichment analysis and KEGG enrichment analysis showed that these genes were significantly enriched in immune pathways, such as cytokine-cytokine receptor, natural killer cell-mediated cytotoxicity, and T-cell receptor signaling pathways. These results suggest that fatty acid metabolism genes play critical roles in modulating the immune response within the tumor microenvironment. We identified three genomic subtypes based on 898 related genes and developed an FMGsScore to comprehensively assess individual breast cancer cases, considering clinical characteristics, tumor mutational load, immune checkpoint expression, and response to immunotherapy. The low FMGsScore group exhibited more abundant immune cell infiltration, better prognosis, higher expression of immune checkpoints, better response to immunotherapy, and higher suitability for immunotherapy. Breast cancer existing literature indicates the significant role of the tumor microenvironment (TME) in the development and progression of breast cancer [25]. Numerous studies have highlighted the crucial involvement of fatty acid metabolism in the TME, which should not be overlooked, and its profound impact on diverse immune cell differentiation and function [26]. Fatty acid synthesis provides the building blocks for effector T-cell function, while fatty acid oxidation inhibits effector T-cell activation and interferon synthesis [27]. Abnormal accumulation of fatty acids in the TME also leads to T-cell senescence and promotes tumor development [28]. In contrast, the low FMGsScore group exhibited higher infiltration abundance of CD8+ and CD4+ T-cells, in line with the expected outcomes of this study. Inhibiting immune checkpoints, such as PD-1 and PD-L1, restores the cytotoxicity of immune cells, achieving an anti-tumor effect [29]. However, the low success rate of immunotherapy for breast cancer is attributed to its low immunogenicity [30]. Certain breast cancer patients, especially those with triple-negative breast cancer (TNBC), exhibit immunogenicity and are associated with poor prognosis and limited response to chemotherapy [31]. These patients represent suitable targets for the immunotherapy we are exploring.

Therefore, this study evaluated the FMGsScore, which was constructed from fatty acid metabolism-related genes, regarding its relevance to immune checkpoints and immunotherapy. We demonstrated a significant association between FMGsScore and immune checkpoint inhibitor (ICI) treatment response in breast cancer. The low FMGsScore group, characterized by higher expression levels of immune checkpoint genes (*PD1*, *PD-L1*, *CTLA4*, *LAG3*, *HAVCR2*, and *IDO1*), exhibited higher ICI scores in *CTLA-4/PD-1* immunotherapy and demonstrated greater sensitivity to ICI treatment compared to the high FMGsScore group. These findings suggest that the use of FMGsScore can assist in identifying breast cancer patients suitable for immunotherapy. Previous research indicates that the amalgamation of fatty acid targets with immunotherapy enhances the efficacy of tumor cell eradication. In a B16 tumor model, concurrent administration of the SREBP1 inhibitor Fatostatin and anti-PD-1 therapy exhibited prolonged survival in mice. This combined approach thwarted tumor growth and extended survival in the B16 mouse model. Furthermore, a reduction in M2-like tumor-associated macrophages (TAMs) and an augmentation in CD8+ T cells were documented [32].

However, most of the current studies involve in vitro cell line experiments or in vivo experiments in mice, which differ significantly from the tumor microenvironment in humans and cannot fully address the immunotherapy challenges in clinical patients. Additionally, single-cell sequencing offers high resolution not achievable with conventional sequencing, enabling measurement of tumor heterogeneity at the level of individual cells. This has great significance for the study of the tumor microenvironment [33]. Addressing the regulatory program of emerging single-cell sequencing multi-omics data remains a challenge in genomics, and the study by Song et al. presents a novel approach, the Single-cell Multi-omics Gene co-regulatory algorithm (SMGR), offering a promising solution. Utilizing single-cell sequencing technology to investigate the metabolic characteristics in the tumor microenvironment allows for better discrimination of metabolic patterns among different cells within the tumor microenvironment [34]. For instance, the study by Yu et al. utilized bulk and single-cell transcriptome profiling to reveal metabolic heterogeneity in human breast cancers, identifying energy-related metabolic signatures to construct a prognostic and therapeutic classifier that distinguished breast cancer patients into two distinct metabolic signature clusters: cluster 1 exhibited high glycolytic activity and lower survival, while cluster 2 was characterized by an enrichment in fatty acid oxidation and glutamine catabolism. This study employed single-cell sequencing technology to provide new metabolic insights, ultimately enabling tailored therapeutic strategies based on patient or cell-type-specific cancer metabolism [35]. Hence, this study employs single-cell sequencing technology to investigate the role of 15 critical fatty acid metabolism genes in cellular communication within the breast cancer tumor microenvironment, aiming to reveal the significance of fatty acid metabolism among immune pathways. The samples were divided into two groups, with high and low scores, using the Addmodulescore function. We observed that antigen-presentation pathways were more active in the high-scoring samples compared to the low-scoring samples. Previous reports have shown that increased lipid accumulation within lipid droplets in tumor-associated dendritic cells (DCs) leads to DC dysfunction due to reduced antigen presentation, resulting in poor stimulation of T-cell responses [36]. Among the 15 selected fatty acid metabolism genes in this study, most were associated with fatty acid oxidation. In the low-scoring samples, lipid accumulation in DC cells led to the inactivation of the antigen-presentation pathway. In high-scoring samples analyzed by CellChat, *LGALS9-HAVCR2*, *LGALS9-CD45*, and *LGALS9-CD44* were observed. *LGALS9* encodes galectin-9, a tandem protein found to interact with *PD-1* and *TIM-3*, regulating T-cell death and serving as a target for cancer immunotherapy [37]. Previous studies have also shown that when *LGALS9* binds to *CD44*, it enhances the inducible regulatory stability and suppressive function of T (iTreg) cells [38]. Therefore, the combination of immune checkpoint inhibitor therapy and inhibition of *LGALS9*, along with fatty acid metabolic pathways, could significantly enhance the success of tumor immunotherapy.

Among the 15 fatty acid-related genes, *NDUFAB1* has the highest prognostic risk factor. Hence, we chose to validate its role through in vitro experiments. MCF-7 and MDA-MB-231 represent different types of breast cancer. MCF-7 belongs to the hormone receptor-positive (ER+) subtype of breast cancer cells, indicating sensitivity to estrogen. On the other hand, MDA-MB-231 represents triple-negative breast cancer (ER-, PR-, HER2-), implying the absence of estrogen and progesterone receptors as well as HER2 expression. Selecting these two cell lines enables a more comprehensive study of various breast cancer types, making them representative choices. Through TCGA database analysis, we observed high expression of *NDUFAB1* in cancer patients. Furthermore, patients with high *NDUFAB1* expression exhibited a lower survival rate. Previous studies have suggested *NDUFAB1* as a potential target for breast cancer. However, its role has not been confirmed through in vivo or in vitro experiments. Through in vitro experiments, we provide the first demonstration of *NDUFAB1* promoting the migration and proliferation of breast cancer cells. This provides additional evidence supporting the involvement of *NDUFAB1* in breast cancer. Previous studies have indicated the role of *NDUFAB1* in cardioprotection [39], but its specific mechanisms in cancer have not been investigated. Our findings reveal an association between *NDUFAB1* and fatty acid elongation, which has been closely linked to various cancers. For instance, it has been demonstrated that extra-long-chain fatty acid protein 5 (*ELOVL5*)-mediated fatty acid elongation promotes the development of gastric cancer [40]. Furthermore, elongation expression of extra-long-chain fatty acid protein 5 (*ELOVL5*) and fatty acid desaturase 1 (*FADS1*) is upregulated in mesenchymal gastric cancer cells (GCs), contributing to the development of ferroptosis [41]. In our study, *NDUFAB1* also emerges as a potential therapeutic target.

## 4. Materials and Methods

### 4.1. Breast Cancer Dataset Source

Gene expression data for BC samples and clinical annotations were gathered using the public TCGA dataset [42] (https://cancergenome.nih.gov/, (accessed on 22 April 2023)). The RNA sequencing data (in FPKM format) were translated to transcripts per million (TPM). The gathered clinical annotations for this project included survival time, survival status, age, sex, stage, grading, and TNM stage. The data were normalized using the “normalize.quantiles” function in the R software (4.3.0) package “preprocessCore” and log2 transformation was applied to unnormalized data. ComBat” algorithm of the “sva” package was used to correct for batch effects caused by non-biotechnical bias. Additional RNA-seq data and clinical survival information for 110 breast cancer samples were obtained from GSE162228 [43] (https://www.ncbi.nlm.nih.gov/geo/, (accessed on 25 April 2023)). The Molecular Taxonomy of Breast Cancer International Consortium (METABRIC) database [44], consisting of 1904 breast cancer samples, was utilized to validate the applicability of 15 Functional Molecular Groups (FMGs) for classification purposes. The accuracy of the model was validated using RNA-seq data and clinical survival information from 327 breast cancer patients in GSE20685 [45]. Single-cell sequence RNA analyses were downloaded from the GEO database (GSE161529) [46]. The expression data were normalized using the R package “Seurat”. The R package “FindVariableGenes” was used to identify the first 2000 highly variable genes. Cellular subpopulation annotation was conducted with the “singleR” package. The R package “t-SNE” was used to map the distribution of cellular components. We obtained information on BRC mutations in the genome, including somatic mutations, copy number variations (CNVs), and analyzed the data using R programs. We obtained breast cancer mutation data, which included somatic mutations and copy number variants (CNVs). Somatic mutation detection and copy number visualization were conducted using the R packages “maftools” and “RCircos”. In order to investigate the predictive validity of FMGsScore for immunotherapy response, we included a cohort of metastatic uroepithelial carcinoma (mUC) (EGAS00001002556) obtained through the R package IMvigor210CoreBiologies (http://research-pub.gene.com/IMvigor210CoreBiologies, (accessed on 29 July 2023)). Additionally, we obtained a cohort of patients treated with anti-PD1 immune checkpoint blockade from the Large Melanoma Genome Sequencing Project (MGSP) [47]. Cellular subpopulation annotation was conducted with the “singleR” package. The R package “t-SNE” was used to map the distribution of cellular components. Single-cell gene expression data from normal patients’ epithelial cells were used as a reference. The R package “inferCNV” (https://github.com/broadinstitute/inferCNV(1.16.0)) was employed to analyze single-cell gene expression data, inferring chromosomal copy number variations in malignant cells. InferCNV was run with default parameters to detect amplifications and deletion events.

### 4.2. Produces Fatty Acid Metabolism-Related Genes Associated with Prognosis

Three gene sets related to fatty acid metabolism, including KEGG fatty acid metabolism pathways, Hallmark fatty acid metabolism genes, and Reactome fatty acid metabolism genes, were obtained from the Molecular Signature Database v7.4 (MSigDB). After removing overlapping genes among the three sets, 309 FMGs were collected. The “limma” package in R was utilized to select 121 differentially expressed genes in TCGA data (logFC = 0.585, FDR = 0.05). Differentially expressed genes (DEGs) were analyzed using univariate Cox regression to identify FMGs associated with overall survival (OS). A total of 15 significant OS-related FMGs were identified individually: *ALOX15B*, *CEL*, *UBE2L6*, *ABCD1*, *HSPH1*, *PTGDS*, *NDUFAB1*, *ALDH3A1*, *ADH6*, *ACSL1*, *PSME1*, *NUDT19*, *ACSL5*, *RDH16*, and *IL4I1*.

### 4.3. Consensus Clustering of FMG Regulators

Unsupervised clustering analysis was employed to identify distinctive patterns of FMG alterations based on the expression of these regulators and breast cancer samples, resulting in significant clusters. Consistent clustering was performed using the R package “Consensus Cluster Plus”, and one thousand repetitions of instances were conducted to ensure the stability of clusters [48].

### 4.4. Gene Set Variation Analysis

The “GSVA” R package was used to investigate differences in biological processes among FMG signature designs [49]. The “ClusterProfiler” package was used for functional annotation, and the gene set file (c2.cp.kegg.v7.2.symbs.GMT) was obtained from the MSigDB database (https://www.gsea-msigdb.org, (accessed on 22 April 2023)) [50,51].

### 4.5. Estimation of TME Cell Infiltration

The relative abundance of each cell infiltrate in the TME was measured using the Single Sample Gene Set Enrichment Analysis (ssGSEA) procedure. Genetic markers for each immune cell type infiltrating the TME were obtained from the Charoentong study, which identified multiple human immune cell subsets, including stimulatory CD8 T-cells, stimulatory dendritic cells, macrophages, NKT cells, and regulatory T-cells [19]. The ssGSEA analysis calculated enrichment values, which were then used to determine the relative abundance of each TME-infiltrating cell in each sample. Additionally, we collected fatty acid-related genes from the Molecular Signature Database v7.4 (MSigDB). The analysis was performed using the R software “GSVA” package (1.48.3) with the parameter method = ‘ssgsea’. Finally, we assessed the correlation between gene expression and pathway scores using Spearman correlation analysis.

### 4.6. Construction of the FMGs’ Gene Signature

We utilized the empirical Bayesian approach of the “limma” R package to construct FMG scoring patterns and identified differentially expressed genes between FMGs’ modification patterns. The significant *p*-values were adjusted using a threshold of *p* < 0.001. Differential analysis and Venn diagrams revealed 898 genes in common across the three FMG clusters. Each gene was subjected to one-way Cox regression analysis to screen for genes associated with prognosis, resulting in a total of 210 genes. Principal component analysis (PCA) was employed to construct the FMG gene signature. *FMGsScore* was defined as:$$FMGsScore=\sum \left(PC1_{i}+PC2_{i}\right)$$
where *i* represents each *FMGs*. *PC*1 and *PC*2 represent the first two principal components obtained from the PCA. In the scRNA-seq dataset, *FMGsScore* is calculated as the mean expression level of FMGs for each individual cell. The single-cell dataset was scored using the “AddModuleScore” function from the R package “Seurat” (4.3.0.1) [52].

### 4.7. Comprehensive Analysis of the FMGsScore Signature with Genomic Mutations, Clinical Information, and Immunity Correlation

We conducted an analysis of genomic mutations, tumor mutation load, and clinical annotations to compare the FMGsScore groups. Subsequently, we employed the Wilcoxon test to evaluate the differences in potential immune checkpoints (e.g., PD-L1, PD-1, CTLA4, LAG3, and TIGHT) between the high- and low-scoring groups. The immune checkpoint inhibitor (ICI) immune epistasis score files were acquired from the Cancer Immunome Database (TCIA, https://tcia.at/home (accessed on 2 May 2023)). The immunophenoscore (IPS) serves as a reliable predictor of CTLA-4 and PD-1 responsiveness and responses to immunotherapy. Moreover, it predicts intergroup differences in the response to immunotherapy with CTLA-4 and PD-1 blockers [53,54].

### 4.8. Analysis of Single-Cell Sequencing Data

Single-cell sequencing data of breast cancer patients and normal breast tissues were downloaded from the GEO database GSE161529. From this dataset, we obtained single-cell sequencing data from 2 TNBC patients, 1 ER+ patient, 1 HR+ patient, and 2 normal individuals. We annotated the single-cell sequencing data using the “SingleR” automated annotation tool with gene markers obtained from CellMarker (http://bio-bigdata.hrbmu.edu.cn/CellMarker/ (accessed on 3 May 2023)) and panglaoDB (https://panglaodb.se/ (accessed on 3 May 2023)). We utilized CellChat (1.6.1) to analyze intercellular communication networks from scRNA-seq data.

### 4.9. Cell Lines Culture and Transfection

The human breast cancer cell line MCF-7 was obtained from Wuhan Pronosai Life Sciences Co., Ltd. (Priscilla, Wuhan, China) and cultured in DMEM (Gibco BRL, Gaithersburg, MD, USA) supplemented with 10% fetal bovine serum (Biological Industries, Cromwell, CT, USA) and 1% penicillin/streptomycin mixture (Liji, Shanghai, China) under conditions of 95% humidity and 5% CO_2_ at 37 °C. The cells were transfected with previously synthesized small interfering RNAs (GenePharma Inc., Shanghai, China) targeting the *NDUFAB1* gene using Lipofectamine3000 (Thermo Fisher Scientific, Waltham, MA, USA) following the manufacturer’s protocol. The siRNA sequences targeting the *NDUFAB1* gene are provided in Appendix A.

### 4.10. Western Blotting

Protein extraction was performed on MCF-7 and MDA-MB-231 cells using whole protein extraction reagents (KGI Biotechnology Ltd., Claremont, CA, USA). Protein quantification was conducted using the BCA method (KGI Biotech Ltd.). The isolated proteins were loaded onto 12% sodium dodecyl sulfate polyacrylamide gels (Solabank Technology Ltd., Shenzhen, China) and subsequently transferred to PVDF membranes (Millipore, Burlington, MA, USA). Subsequently, the PVDF membranes were blocked with 5% non-fat milk for 2 h and then incubated overnight at 4 °C with primary antibodies at a dilution of 1:1000. The primary antibodies used were *GAPDH* (Affinity Biosciences Ltd., Changzhou, China) and *NDUFAB1* (YN2973, Immunoway, Plano, TX, USA). After washing, the membranes were incubated with secondary antibodies (Proteintech, Rosemont, IL, USA) at a dilution of 1:5000 for 2 h at room temperature. Following the incubation with secondary antibodies, the membranes were exposed to ECL developer (Epizyme Biomedical Technology Co., Ltd., Suzhou, Jiangshu, China) for 30 s, and the protein bands were visualized using the Molecular Imager ChemiDoc XRS system (BIO-RAD, Hercules, CA, USA). The protein bands on the Western blot were then analyzed using ImageJ software (Java 1.8.0_345(64-bit)).

### 4.11. qRT-PCR

Total RNA was extracted from the cell lines using AG RNAex Pro RNA Kit (AG21101, Accurate Biotechnology, Changsha, Hunan, China). Subsequently, cDNA was synthesized and reverse transcribed using the Evo M-MLV RT Kit with gDNA Clean for RT-qPCR (AG11705, Accurate Biotechnology, Changsha, Hunan, China). Real-time polymerase chain reaction (RT-PCR) was performed with the SYBR Green Pro Taq HS premixed qPCR kit (AG11701, Accurate Biotechnology, Changsha, Hunan, China), and expression levels were calculated using the 2^^(−ΔΔCt)^ method. mRNA expression was normalized to the expression level of β-actin mRNA. All primers were provided by Accurate Biotechnology Co., Ltd. (Changsha, Hunan, China), and the detailed primer sequences are shown in Appendix A. All data are presented as the mean ± SD of three independent experiments. * *p* < 0.05, ** *p* < 0.01, *** *p* < 0.001, **** *p* < 0.0001.

### 4.12. CCK8 Assay to Detect Cell Proliferation

Cell Counting Kit-8 (CCK-8) kit (bs-4975R, Bioss, Beijing, China) was used to assess MCF-7 cell viability. After transfection, the cells were incubated for 7–8 h, digested, and resuspended with fresh medium. The cell density was then adjusted to 2 × 104 cells/mL and 0.1 mL of the cell suspension was added to each well in a 96-well plate, with 5 wells per group (NC, siNC, and si*NDUFAB1*). The cells were incubated at 37 °C with 5% CO_2_ for 12, 24, 48, and 72 h. Afterward, 10 μL of CCK-8 solution was added to each well and incubated for 90 min at 37 °C in a cell incubator. The absorbance values were measured at 450 nm using a Spectra MR ELISA (Dynex, Charlottesville, VA, USA). All data are presented as the mean ± SD of three independent experiments. * *p* < 0.05, ** *p* < 0.01, *** *p* < 0.001, **** *p* < 0.0001.

### 4.13. Healing Assay

MCF-7 cells were transfected in each group for 7 h, digested, and spread across a monolayer in a six-well plate. The cells were seeded at a density of 4.0 × 10^5^ cells/mL, with 2 mL of cells added to each well, and incubated for 24 h to ensure even distribution. Vertical lines were drawn in advance on the plate, and each well was scored 3 times using a 200 μL pipette tip, spaced equally and parallel to each other. After the scratching, cells were gently washed 3 times with 1 mL of PBS, and then 2 mL of complete medium was added to each well. The cells were observed under an inverted microscope and recorded at 0 h. After 24 h, the record was re-observed under the microscope, and the widths of the scratches at 0 and 24 h were calculated using software and analyzed with Image J software(Java 1.8.0_345(64-bit)). All data are presented as the mean ± SD of three independent experiments. * *p* < 0.05, ** *p* < 0.01, *** *p* < 0.001.

### 4.14. Statistical Analysis

Pearson correlation analysis detected correlations between variables. Two continuous variables that fit a normal distribution were compared using a *t*-test. The Kruskal–Wallis test was used to compare differences between two groups or more. The optimal cutoff value was obtained by the “surv_cutpoint” function in the “survminer” R package, based on the correlation between the survival outcome and the FMGsScore for each individual dataset. The samples were divided into two groups, a high FMGsScore group and a low FMGsScore group. The cutoff value for the training set is -12.63661. Survival curves were generated for each subgroup in the dataset using the Kaplan–Meier method, and statistical differences were determined using the log-rank test. All statistical analyses were performed using R4.3.0 (https://www.r-project.org/). *p* values are two-sided. A *p* value of less than 0.05 was considered statistically significant.

## 5. Conclusions

This study utilized FMGs to classify breast cancer subtypes, identifying the significant therapeutic potential of fatty acid metabolism-related genes in the tumor microenvironment (TME) and breast cancer (BC). This research contributes to the advancement of novel immunotherapeutic strategies.

## Figures and Tables

**Figure 1 ijms-24-13209-f001:**
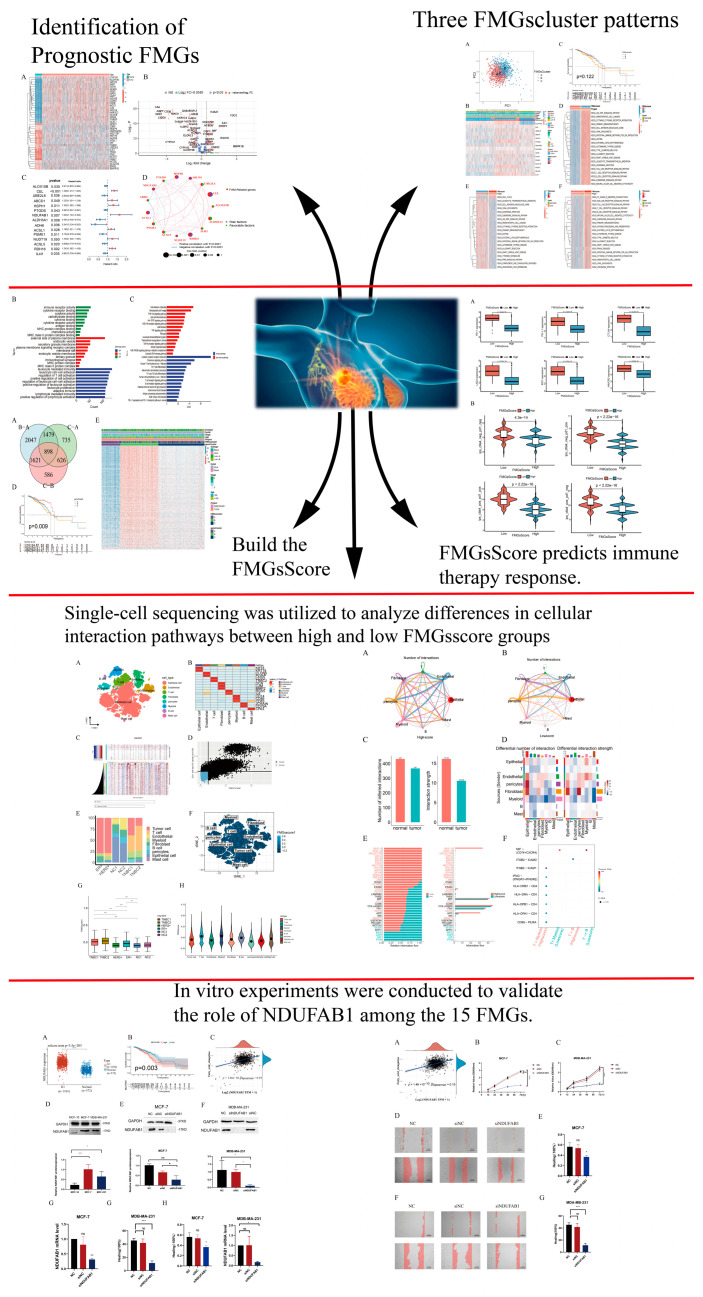
The flowchart of this study.

**Figure 2 ijms-24-13209-f002:**
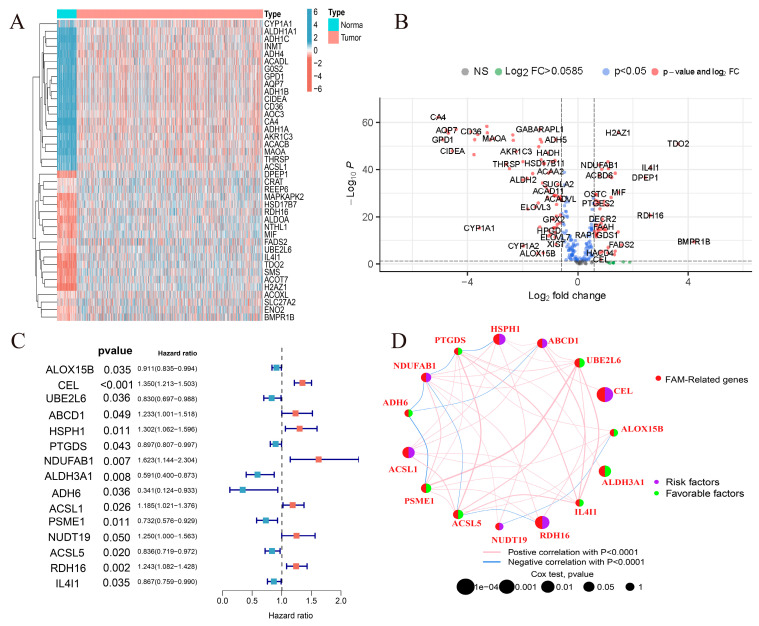
Identification of prognostic FMGs in BC patients. (**A**) Heatmap illustrating the differential expression of 40 genes from the FMGs. (**B**) Volcano plot exhibiting 121 DEGs among FMGs. (**C**) The 15 FMGs associated with prognosis. (**D**) The interaction between FMGs in breast cancer. The circle size represented the effect of each gene on the prognosis, and the range of values calculated from Log-rank test was *p* < 0.001, *p* < 0.01, *p* < 0.05, and *p* < 0.1, respectively. Purple in the right part of the circle, risk factors of prognosis; green in the right part of the circle, protective factors of prognosis. The lines linking regulators show their interactions, and thickness shows the correlation strength between regulators. The negative correlation is marked with blue and the positive correlation with red.

**Figure 3 ijms-24-13209-f003:**
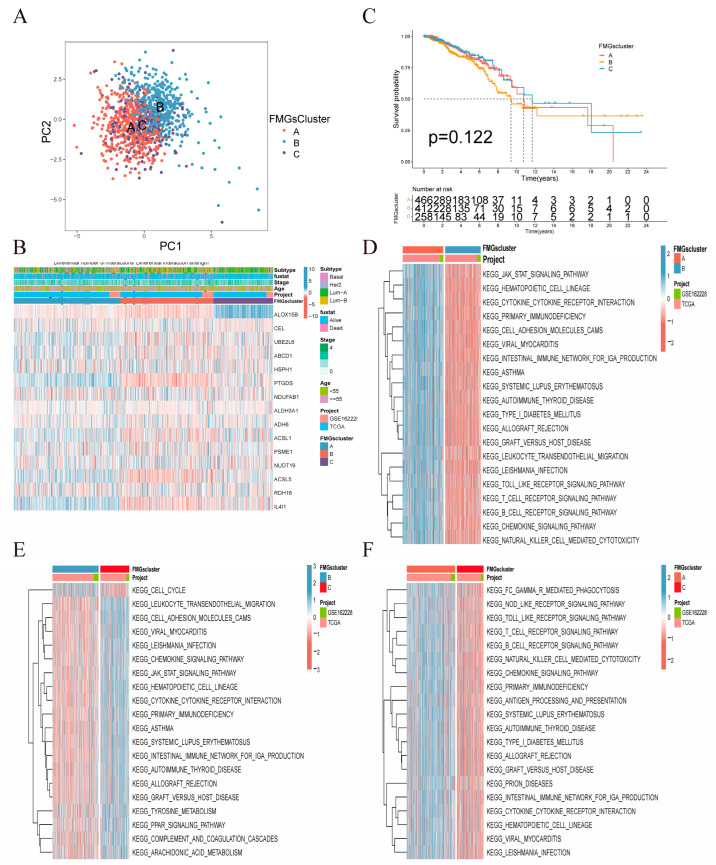
Mediation patterns of 15 fatty acid-related genes in breast cancer. (**A**) Principal component analysis (PCA) analysis of the three FMGsCluster modification patterns. (**B**) Heat map of the clinical relevance of the three fatty acid subtypes. (**C**) Overall survival of the fatty acid modification patterns was determined using Kaplan–Meier curves. (**D**–**F**) GSVA enrichment analysis shows the activation status of biological pathways under different fatty acid modification patterns. Heat maps were used to visualize these biological processes, with red representing activated pathways and blue representing inhibited pathways.

**Figure 4 ijms-24-13209-f004:**
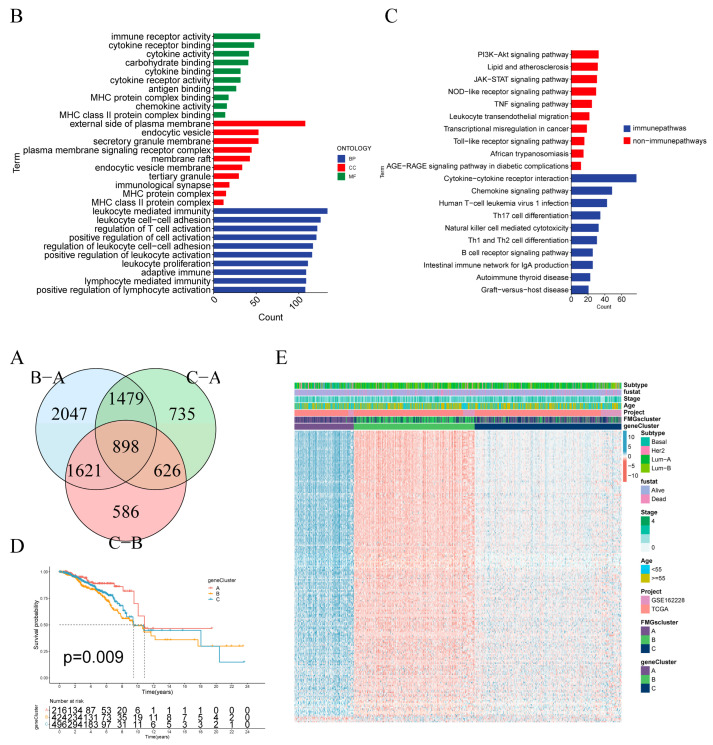
Construction of gene signatures based on differential genes of FMGsCluster. (**A**) The Venn diagram illustrates the shared genes among three FMGsCluster subtypes. (**B**,**C**) GO and KEGG enrichment analysis of 898 genes. (**D**) Kaplan–Meier curves were used to demonstrate the overall survival of the three gene clusters. KM curves showed that the FMGs genome was significantly associated with the difference in the overall survival of BC patients. (**E**) Clinical relevance of FMGsCluster and gene cluster plotted on a heat map. Annotation of BC patients was based on the gene clusters, FMG clusters, stage of the tumor, survival status, and age. CC: cellular components (CC); MF: molecular functions (MF); BP: biological processes. Immune Pathways: Pathways related to organismal immune responses in the KEGG database. Non-Immune Pathways: Enriched pathways in the KEGG database that are unrelated to organismal immune responses.

**Figure 5 ijms-24-13209-f005:**
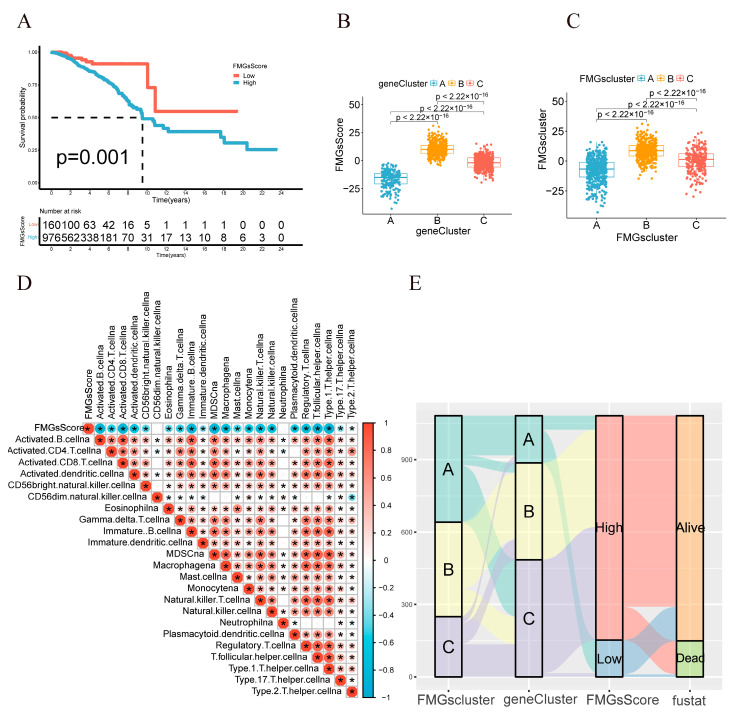
Construction of FMGsScore according to intersect gene. (**A**) K–M survival curves between groups with high and low FMGsScore scores. (**B**,**C**) Differences in FMGsScore between different FMGsClusters and geneClusters. (**D**) Correlation of FMGsScore with immune cell infiltration by Spearman correlation analysis. (**E**) Alluvial illustration indicating FMGsCluster, geneCluster, FMGsScore and fustat changes. * *p* < 0.05 was considered statistically significant.

**Figure 6 ijms-24-13209-f006:**
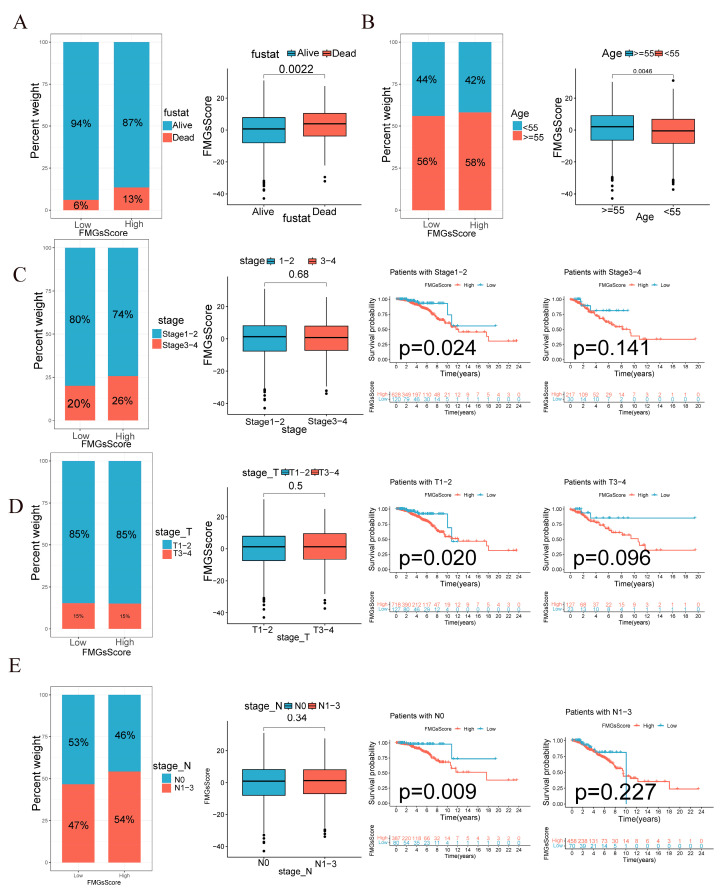
FMGsScore and clinical relevance in the TCGA-BC cohort. The ratio of fustat (**A**), age (**B**), stage (**C**), T (**D**), and N (**E**) between high and low FMGsScore patients. Patients are staged according to the TNM system (tumor-node-metastasis), with T representing the size of the tumor itself or the extent of invasion, and N representing invasion.

**Figure 7 ijms-24-13209-f007:**
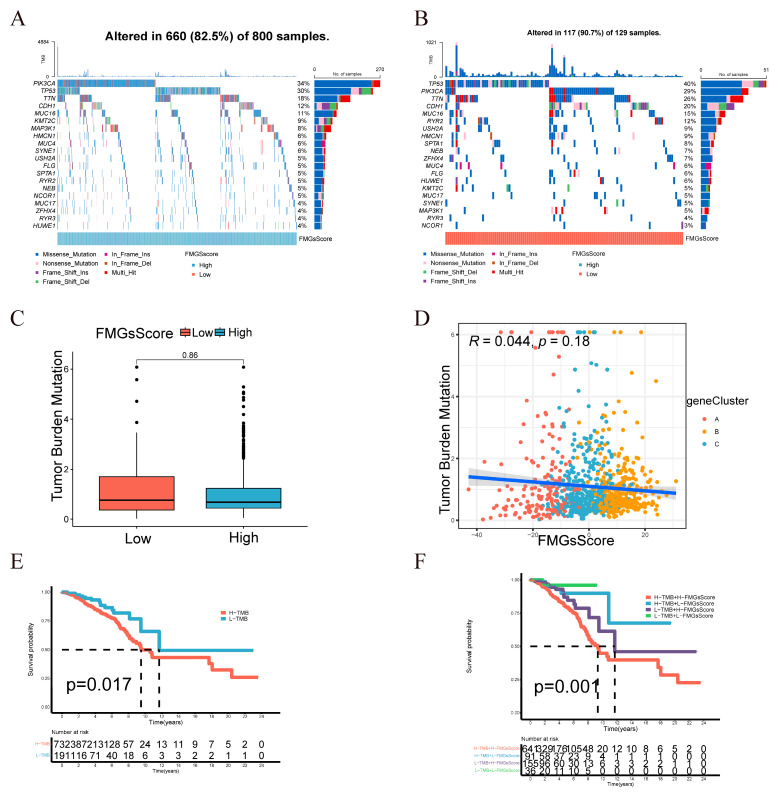
Characterization of tumor somatic mutations across FMGsScore groups. (**A**) Waterfall plots of somatic mutations in the high FMGsScore group tumors. (**B**) Waterfall plots of somatic mutations in the low FMGsScore group tumors. (**C**) The box plots illustrate variations in tumor mutation burden (TMB) between the high and low FMGsScore subgroups. (**D**) Positive correlation between FMGsScore and TMB. (**E**) Patient survival analysis in low and high TMB groups using Kaplan–Meier curves. (**F**) Overall patient survival stratified by FMGsScore and TMB using Kaplan–Meier curves. TMB: the tumor mutation burden.

**Figure 8 ijms-24-13209-f008:**
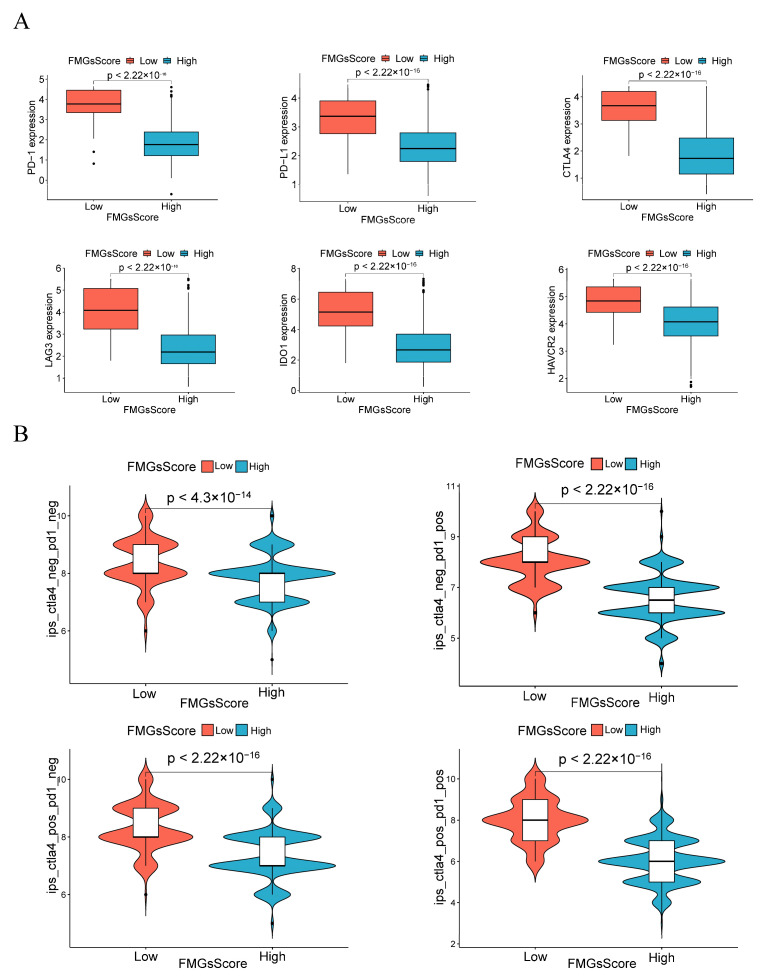
Effect of FMGsScore in immunotherapy. (**A**) Differences in immune checkpoint gene expression between high and low FMGsScore groups. (**B**) Immunotherapy response between high and low FMGsScore subgroups.

**Figure 9 ijms-24-13209-f009:**
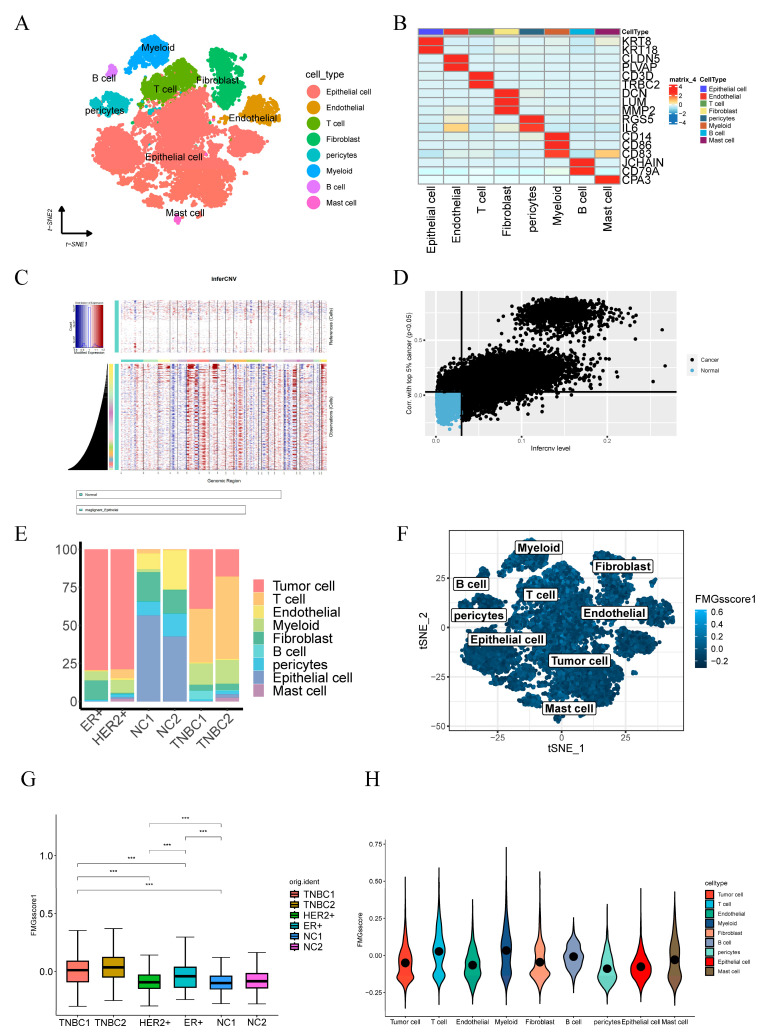
Atlas of single-cell sequencing in six breast cancer patients. (**A**) t-SNE plot showing cell subtypes in breast cancer samples. (**B**) Gene heat map annotated with single-cell sequencing data. (**C**) The heatmap illustrates extensive CNV in tumor epithelial cells, with red denoting amplifications and blue indicating deletions. (**D**) Tumor and normal epithelial cells were distinguished based on their CNV score ratings. (**E**) Stacked plots showing the cellular composition between each sample. (**F**) The 15 fatty acid metabolism gene scores for the eight major cell types are represented by t-SNE plots. (**G**) Differences in fatty acid metabolism gene scores between the six samples are shown in violin plots (*** *p* < 0.001). (**H**) Differences in the scoring of fatty acid metabolism genes among eight cell types.

**Figure 10 ijms-24-13209-f010:**
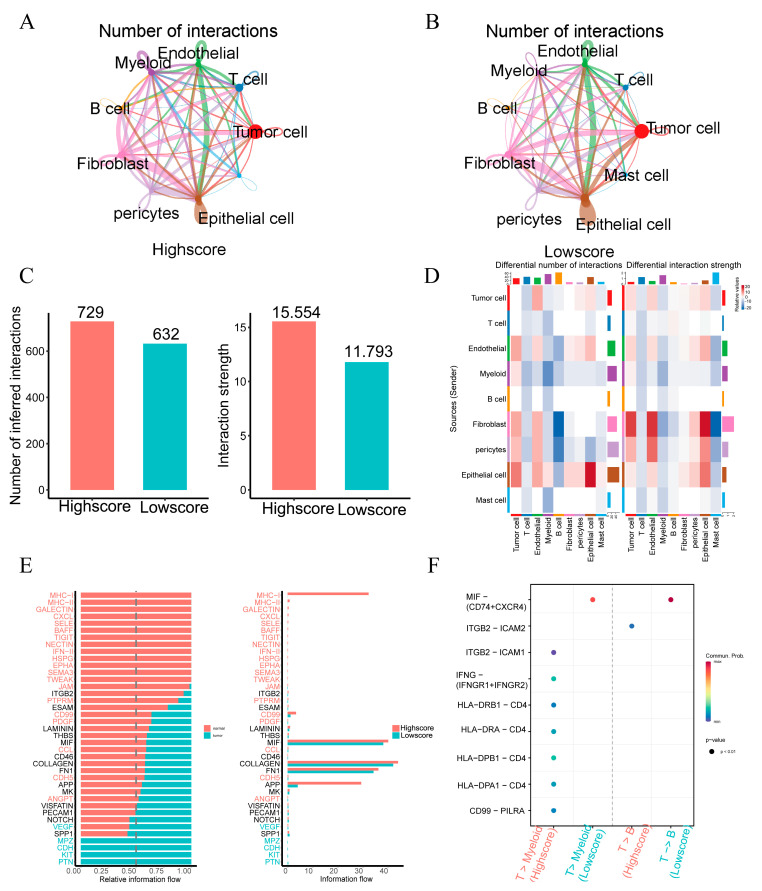
Differences in intercellular interactions between high- and low-scoring groups. (**A**) Cell interactions network diagram for Highscore grouping of cell communication numbers. (**B**) Cell interactions network diagram for Lowscore grouping of cell communication numbers. (**C**) Comparison of the number and strength of interactions between Highscore and Lowscore. (**D**) The number and strength of cellular interactions between the Low-score and High-score groups were compared. Deeper shades of red indicate more active cell communication within the Low-score group, while deeper shades of blue indicate more active cell communication within the High-score group. (**E**) Differences in cellular communication pathways between the Lowscore and Highscore groups. (**F**) Comparison of important ligand-receptor pairs for T-cell signaling to epithelial and myeloid cells. The color of the dots reflects the communication probability, and the size of the dots indicates the calculated *p*-value. An empty space indicates a zero communication probability.

**Figure 11 ijms-24-13209-f011:**
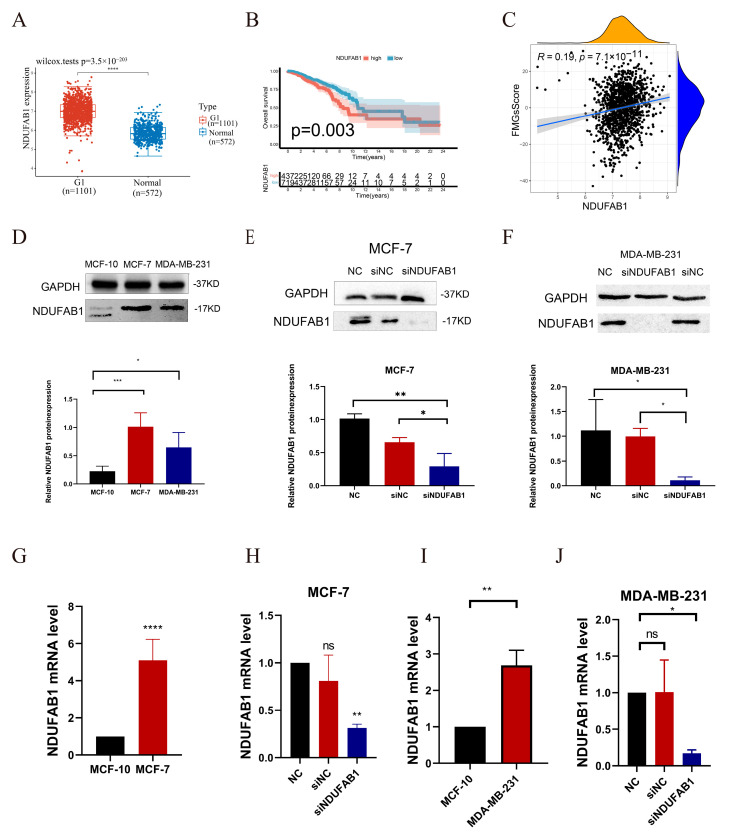
Validation of the expression of *NDUFAB1* in breast cancer. (**A**) Differential expression of *NDUFAB1* in TCGA data. (**B**) Analysis of the relationship between *NDUFAB1* and OS in breast cancer patients. (**C**) The gene expression level of *NDUFAB1* is positively correlated with the FMGsScore. (**D**) The Western blot was performed to assess the relative expression difference of the *NDUFAB1* protein between breast cancer cell lines (MCF-7 and MDA-MB-231) and the normal breast epithelial cell line (MCF-10). (**E**,**F**) The Western blot was performed to assess the relative expression d of the *NDUFAB1* protein after 2 days of transfection. (**G**) qRT-PCR was performed to assess the expression difference of *NDUFAB1* between breast cancer cell lines (MCF-7) and the normal breast epithelial cell line (MCF-10). (**H**) The qRT-PCR assessment of *NDUFAB1* mRNA levels after 2 days of transfection. (**I**) qRT-PCR was performed to assess the expression difference of *NDUFAB1* between breast cancer cell lines (MDA-MB-231) and the normal breast epithelial cell line (MCF-10). (**J**) The qRT-PCR assessment of *NDUFAB1* mRNA levels after 2 days of transfection. All data are mean ± SD of three independent experiments. * *p* < 0.05, ** *p* < 0.01, *** *p* < 0.001, **** *p* < 0.0001, ns: not statistically significant.

**Figure 12 ijms-24-13209-f012:**
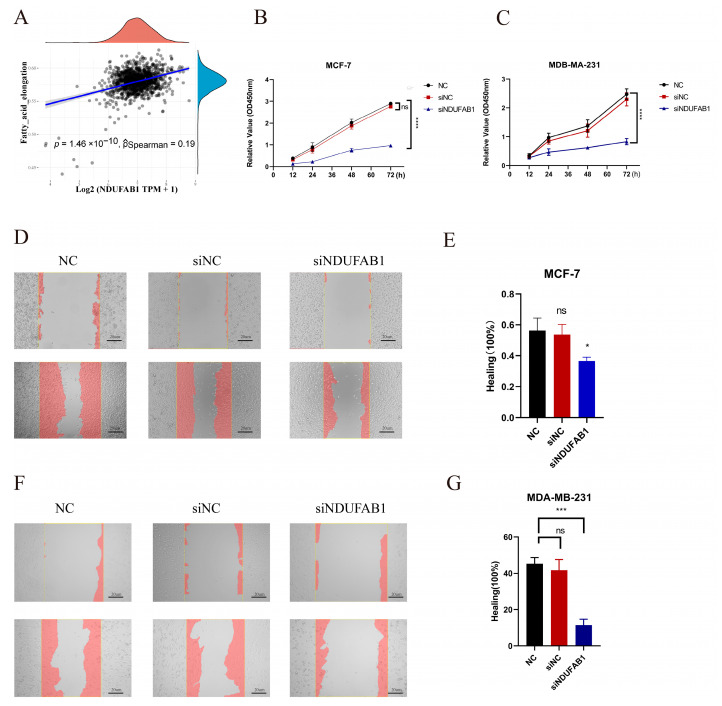
(**A**) Assessment of the correlation between *NDUFAB1* and fatty acid elongation. (**B**,**C**) CCK8 assay. Cell viability was significantly decreased after *NDUFAB1* was knocked down in MCF-7 and MDA-MB-231 cells. (**D**–**G**) Scratch-wound healing assay. Significantly slower wound healing was observed in MCF-7 and MDA-MB-231 cells with knockdown of *NDUFAB1*. All data are mean ± SD of three independent experiments. * *p* < 0.05, *** *p* < 0.001, **** *p* < 0.0001, ns: not statistically significant.

## Data Availability

The public TCGA dataset (https://cancergenome.nih.gov/, (accessed on 22 April 2023)) was used to gather gene expression data for BC samples and clinical annotations. Additional RNA-seq data and clinical survival information for 110 breast cancer samples were obtained from GSE162228 (https://www.ncbi.nlm.nih.gov/geo/, (accessed on 25 April 2023)). RNA-seq data and clinical survival information of 327 breast cancer patients from GSE20685 were used to validate the accuracy of the model. Data for single-cell sequence RNA analysis were downloaded from the GEO database (GSE161529). we included a metastatic uroepithelial carcinoma (mUC) cohort (EGAS00001002556), obtained through the R package “IMvigor210CoreBiologies” (http://research-pub.gene.com/IMvigor210CoreBiologies, (accessed on 29 July 2023)).

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
