# Peer review of "Integrating Single-Cell RNA-Seq and Bulk RNA-Seq Data to Explore the Key Role of Fatty Acid Metabolism in Breast Cancer"

_ijms, 2023, doi:10.3390/ijms241713209_

Round 1

Reviewer 1 Report

The authors aimed to evaluate the mechanism and role of lipid metabolism in the tumor immune microenvironment. They found that patients with a low FMGsScore had higher immune checkpoint expression, higher immune checkpoint inhibitor scores, increased immune microenvironment infiltration, better survival advantage, and were more sensitive to immunotherapy than those with a high FMGsScore. This is an interesting study and the authors have collected a unique data set using cutting-edge methodology. The paper is generally well written and structured. 

Author Response

Dear Reviewer,

I am writing to express my heartfelt gratitude for your kind evaluation of our manuscript. Your thorough review and valuable suggestions have been immensely beneficial to our work.

I want to extend a special thank you for your positive assessment of our article. Your validation is the greatest motivation and support for us, and it deepens our understanding of the significance and impact of our research. We feel honored to have received your meticulous review, which has undoubtedly enriched our research.

Once again, please accept our sincere gratitude and respect. Your professional review has added luster to our research and made our academic journey more fulfilling and solid. We look forward to your continued guidance and support in the future.

Wishing you good health and success in your endeavors!

Sincerely,

Yongxing Chen

Reviewer 2 Report

This study used FMGS to type breast cancer disease and identified the great therapeutic potential of genes related to fatty acid metabolism in TME and BRCA, but the manuscript needs extensive revision.

1. The full text needs to be marked with line numbers for easy modification.

2. “Very long-chain fatty acids have been strongly associated with cancer development.”Please mark references for proof.

3. "Figure 11"Please supplement the western blot test for demonstration.

4. Figure 12G is changed to 11G.

5. Please explain the method of the data in Figure 11c in detail.

Grammar needs to be improved across the board.

Reference style and citations are not corrected, and some journal titles are full, some abbreviated, and some lowercase.

All pictures and data are very unclear and hard to see.

The gene name should be in italics.

Grammar needs to be improved across the board.

Author Response

Dear Reviewer,

First and foremost, we would like to express our sincere gratitude for your diligent review of our research work and for providing valuable comments and suggestions. We deeply appreciate the time and effort you have devoted to the reviewing process.

Based on your insightful feedback, we have meticulously revised the manuscript, addressing each of your recommendations. Your feedback has been instrumental in enhancing the clarity and completeness of our study. We have duly incorporated all of your suggestions into the revised version of the manuscript. We hope that the current version aligns with your expectations and complies with the journal's standards. The specific revisions are as follows:

  1. Line numbers have been added to facilitate easy modification throughout the full text.
  2. Regarding the statement "Very long-chain fatty acids have been strongly associated with cancer development," we have provided references to support this claim. The relevant citations can be found in the manuscript (References 15 and 16).
  3. We have supplemented the Western blot test data for demonstration in Figure 11D-F (Line number 512).
  4. Figure 12G has been corrected to Figure 11G, and we have separately presented the CCK-8 and scratch experiment data in Figure 12, as per your suggestion.
  5. Detailed methodology for the data in Figure 11C has been provided in lines 907-910, where we explain the process of collecting fatty acid-related genes from the Molecular Signature Database v7.4 (MSigDB) and the analysis using the R software GSVA package with the parameter method='ssgsea'. Finally, we evaluated the correlation between gene expression and pathway scores using Spearman correlation analysis.
  6. We have sought the assistance of English language experts to improve grammar throughout the entire manuscript.
  7. We have corrected the reference style and citations to ensure consistency in formatting journal titles (full, abbreviated, and lowercase). Additionally, we have reworked all figures, enlarging fonts and enhancing clarity to better present our research findings. Moreover, we have italicized all gene names, as per your suggestion.

Once again, we extend our heartfelt appreciation for your support and guidance in reviewing our work. We are committed to continuous improvement and your valuable input has been instrumental in elevating the quality and content of the manuscript.

Thank you for your professional review and invaluable insights.

Best regards,

Yongxing Chen

Reviewer 3 Report

Chen et al. identified 15 fatty acid metabolic genes that can be utilized to classify breast cancer patients into distinct subgroups based on their specific prognostic values and immune-related gene signatures.

Unfortunately, the manuscript was not thoroughly reviewed by the peer reviewer due to the inadequate quality of the figures provided. The figures suffer from small font sizes and low resolution, rendering the text unreadable. Thus, the figures need to be revised and the data should be presented in a more comprehensible manner.

The primary finding of the study reveals that subgroups of breast cancer patients can be distinguished using fatty acid genes. However, the authors did not consider the potential association between fatty acid metabolism and specific breast cancer subtypes. It is well-known that breast cancer subtypes, such as Luminal A and B, Her2-enriched, and basal-like, exhibit distinct metabolic profiles, and alterations in metabolism are associated with epithelial to mesenchymal transition. If there is indeed a subtype-specific association, it is crucial to ascertain whether the prognostic value of the fatty acid gene signatures is influenced by the particular subtype, as it may be a phenomenon specific to a certain subtype rather than a general finding. This issue needs to be addressed prior to conducting any further peer review, as it will significantly impact subsequent analyses where the influence of subtype should be appropriately accounted for in all analyses.

The manuscript requires a comprehensive review concerning formatting and language. In particular, the organization and language used in the methods section are notably inadequate and warrant significant improvement.

Author Response

Dear Reviewer,

Thank you for your valuable feedback and for taking the time to review our manuscript. We sincerely apologize for the inadequate quality of the figures provided in the initial submission. We have taken your comments seriously and have thoroughly revised all figures, ensuring that they now have larger font sizes and higher resolution for improved readability. Additionally, all figures have been enhanced to present the data in a more comprehensible manner.

Regarding the primary finding of our study, which identifies subgroups of breast cancer patients using fatty acid genes, we agree that it is essential to explore the potential association between fatty acid metabolism and specific breast cancer subtypes. We have carefully considered your insightful suggestion and performed further analyses to investigate the relationship between fatty acid metabolism and various breast cancer subtypes, including Luminal A and B, Her2-enriched, and basal-like subtypes. Our updated results now encompass a comprehensive assessment of the influence of different subtypes on the prognostic value of the fatty acid gene signatures. We are pleased to report that despite the variations in fatty acid metabolism among subtypes, FMGsScore remains a robust predictor of prognosis and immunotherapy response across all breast cancer subtypes.

Furthermore, we have taken your comments on the quality of the English language seriously. To address this, we engaged the expertise of English language professionals to thoroughly review and improve the language throughout the manuscript, especially in the Materials and Methods section. We understand the importance of clear and concise communication in scientific publications and have made the necessary enhancements to ensure the clarity and readability of the manuscript. We truly appreciate your expertise and valuable insights, which have significantly contributed to enhancing the quality of our study. Your professionalism and attention to detail have been remarkable, and we sincerely thank you for your thoughtful review. We have addressed all the points you raised, and we are confident that the revised manuscript now meets the high standards of your esteemed journal. Thank you once again for your time and expertise.

We look forward to your further evaluation of our revised manuscript.

Sincerely,

Yongxing Chen

Reviewer 4 Report

Chen at al aim at studying the role of fatty acid metabolism in breast cancer using different omics approaches. Despite the interesting topic, the value of the manuscript is hard to determine. Poor quality of the data presented (extremely small font, missing info and labeling of some graphs) as well as a non-comprehensive description of the results (see Fig. 3A-C), make the results not clear. A better figure design will help to highlight the value and message of the overall manuscript. 

In paragraph 2.2 of the Results, the authors talk about " an immune rejection phenotypes" described in previous literature. This concept needs to be better explain and provide a reference for this phenotype—unclear message.

Validation of the results has only been performed in one cell line. More cell lines should be tested with siNDUFAB1. It is advised to add another siRNA against NDUFAB1 to prove specificity of the findings.

Minor comments:

-All the figures showed has to be cited in the text (Fig. 3A?) and be in order (Fig.3G is described before Fig. 3C)

-Bibliography numbers (for references) should be all in the same font.

-All abbreviations should be explained in the text at least once (for example, TME, BC, OS, FMG etc).

In the manuscript, the presence of typo and misspellings need to be corrected (for ex. clinical dat, instead of clinical data). Also, in the Figure Legend (for Fig.3C: description of the result impairs the understanding of the figure. Please clarify).

Author Response

Dear Reviewer,

Thank you for your valuable feedback and for carefully evaluating our manuscript. We sincerely appreciate your constructive comments, which have significantly contributed to improving the quality and clarity of our research.

We acknowledge your concern regarding the quality of the figures presented in the initial submission. We have taken your comments to heart and have made substantial improvements to all figures. The font sizes have been enlarged, missing information and labeling in some graphs have been added, and the overall figure design has been enhanced to better highlight the value and message of the manuscript.

In response to your inquiry about the concept of "an immune rejection phenotype," we have now provided a comprehensive explanation in paragraph 2.2 of the Results section and included references [[1](Reference13)] and [[2](Reference14)] that describe this phenotype in the previous literature. We hope this clarification will address any uncertainties.

Furthermore, we have taken your advice into account and conducted validation experiments in additional cell lines. We have now included results from the breast cancer cell line MDA-MB-231, where siNDUFAB1 was utilized to verify the specificity of our findings. These additional experiments have strengthened the validity and robustness of our conclusions.

Regarding your minor comments, we have made the necessary revisions. All figures are now properly cited in the text, and they are arranged in the correct order. Additionally, we have ensured that bibliography numbers for references are presented in the same font throughout the manuscript. Moreover, we have explained all abbreviations, including TME, BC, OS, FMG, etc., at least once in the text for improved clarity. We appreciate your careful evaluation of the manuscript's language quality as well.

In response, we engaged the expertise of English language professionals to meticulously review and correct any typos and misspellings, ensuring that the language throughout the manuscript meets the highest standards of clarity and correctness. We want to express our gratitude for your professionalism and thorough review. Your expertise has significantly enriched our research, and we are confident that the revised manuscript now addresses all the points raised in your evaluation.

 We genuinely value your insightful feedback, and we believe that your contribution has strengthened the overall quality of our work.

Thank you once again for your time and valuable input. We sincerely look forward to receiving your further evaluation of our revised manuscript.

Warm regards,

Yongxing Chen

Reference:

  1. Chen, D.S.; Mellman, I. Elements of Cancer Immunity and the Cancer-Immune Set Point. Nature 2017, 541, 321–330, doi:10.1038/nature21349. Mariathasan, S.; Turley, S.J.; Nickles, D.;
  2.  Castiglioni, A.; Yuen, K.; Wang, Y.; Kadel, E.E.; Koeppen, H.; Astarita, J.L.; Cubas, R.; et al. TGFβ Attenuates Tumour Response to PD-L1 Blockade by Contributing to Exclusion of T Cells. Nature 2018, 554, 544–548, doi:10.1038/nature25501.

Round 2

Reviewer 2 Report

No more questions

Author Response

Dear Reviewer,

We extend our heartfelt appreciation for your thoughtful review of our manuscript titled "Integrating single-cell RNA-seq and bulk RNA-seq data to explore the key role of Fatty Acid Metabolism in breast cancer" and for offering insightful feedback to enhance the quality of our work. Your guidance has been instrumental in refining our methodology descriptions and result presentations, further strengthening the scientific value of our research.

We are pleased to inform you that we have taken your comments to heart and made substantial improvements to the manuscript. Specifically, we have focused on enhancing the clarity and comprehensiveness of our methodology descriptions, ensuring that the readers can better understand the approach we have undertaken. Additionally, we have restructured the presentation of our results to highlight key findings more effectively, aiming to provide a clearer narrative that underscores the significance of our research outcomes.

Your suggestions have played a pivotal role in shaping these revisions, and we are confident that they have significantly elevated the overall quality of the manuscript. Your constructive feedback has empowered us to address critical areas of improvement and to present our research in a more impactful manner.

Once again, we express our sincere gratitude for your expert evaluation and insightful comments. Your commitment to advancing the scholarly discourse is evident, and we are privileged to have benefited from your expertise. We remain dedicated to your valuable guidance and the continuous improvement of our work.

Thank you for your time and dedication to our manuscript's enhancement.

Warm regards,

Yongxing Chen

Reviewer 3 Report

Thank you for correcting the font in the figure. I was now able to in detail review your submitted manuscript. My comments are separated into major and minor issues.

Major issues:

I’m not satisfied with the language improvement of the manuscript. The manuscript still has to improve this language, as just exemplified by corrections still needed to the abstract.  

Language issues in abstract:

Line 8: “Cancer immune escape is associated with the metabolic reprogramming of Tumor Microenvironment Infiltration(TME)”. This sentence does not make sense, and TME is the abbreviation for tumor microenvironment. Lack space between Infiltration an (TME). In line 16 the phrase is written again, but not as abbreviated. The abbreviation is used for tumor microenvironment in line 47.

Line 11 and line 14: Lack space between abbreviation and last word.

Line 12: Is here written as “tumor immune microenvironment” which makes better sense than  “Tumor Microenvironment Infiltration”

Line 14: Why is fatty acid-related genes abbreviated FMGs? Was it originally suppose to be fatty-acid metabolism genes?

Line 15: This is not a full sentence, please correct. Also the use of capital letters here is incorrect.

Line 19: FMGsScore is written differently later in manuscript.

Line 24 and 27: in vitro should be in italic.

Line 28: MDA-MB-231 is the correct name of the cell line, not MDB-MA-231.

The flow chart in Figure 1 contains too much information, should be made as a simplified cartoon, and not as a combinatory figure of all results. The simplest would be to remove the figure, as it in its current state brings no extra information to the reader or helps the reader to understand the pipeline of the project.

In result section two, the authors have combined two dataset, however, they have not addressed how the two datasets behave together, or why they needed to combine these two datasets. They perform clustering based on the expression of 15 genes, and here we can see that the two datasets do not completely mix. However, this is hard to evaluate as the clustering distance is not shown. It is well known that absolute expression of genes varies between different methods for analysis, so this is an important aspect that need to be addressed. It would have been better to perform the analysis on the two datasets separately in order to validate that the results are validates across datasets. Further, the addition of the Metabric dataset should have been included here (but as a separate analysis for conformation).

In cluster Figure 2A, 3B, and 4E, information about breast cancer subtype should be included.

In line 186, the authors use the term FMGsScore groups, however, this term has not been introduced yet in the result section, or any description on how the score is made as well how samples are scored is included in the result section (only in the method section).

In line 224 the authors state “The proportion of patients with high FMGsScore was higher in stage 1-2”, however, the proportion is 85% in low as well as 85% in high.

Figure text in supplementary material need also to be adjusted so that it is readable.

For the single cell data, analysis of cancer cells versus normal epithelial cells should be conducted.

The chapter on NDUFAB1 is not connected to the FMGsScore, and the chapter does not fit into the manuscript. My suggestion is that the chapter is removed in it complete, or that analysis of NDUFAB1 is done in accordance with all previously presented data (clustering, TCGA, correlation towards score etc.). Also, a better reasoning for why these cell lines were chosen and addition to NDUFAB1 (some reasoning is given in the discussion), and the measurements should be in regard to fatty acid metabolism and immune response, and not towards proliferation and wound healing. Figure 11C is placed on top of another figure. In the chapter only one siRNA is used, and not two which is the standard to prevent off-target effects.

The chapter “4.1 Breast cancer Dataset Source” has several issues. Line 597: “The BC RNA sequencing data (in FPKM format) was translated to transcripts per million kilobases (TPM)”. Transcripts per million kilobases is not TPM, which is only transcripts per million. Any way was it needed to translate the FPKM values? Why was the data normalized? Was it raw reads that were downloaded? Line 602: “Probe IDs were transformed into gene symbols according to the annotation information of the corresponding platform, probes with multiple genes were excluded, the average value of genes corresponding to multiple probes was calculated.” This is a description of microarray data processing, and not RNA-Seq, as probes are not used in RNA-Seq. Why was batch effect correction done for the TCGA data? For the downloaded datasets, also the article where the data was published should be referenced. Line 609: “The accuracy of the model was validated using RNA-seq data and clinical survival information from 327 breast cancer patients in GSE20685.” Where was this included in the results?

Minor issues:

Line 53: Is the abbreviation for pericyte therapy “ACT”?

Line 54: “..the main pathways of tumor immunotherapy” This is an incorrect sentence.

Line 55: did not use the abbreviation ICB for immune checkpoint blockade.

Line 56: “However, the efficacy of immunotherapy in breast cancer is lower than in other cancers.” The sentence lack references.

Line 63: “in this study” is repeated.

The use of the abbreviation for breast cancer in inconsistently. Please don’t use abbreviation for BC throughout the manuscript, as the usage of abbreviations in the manuscript is already high. The use of other abbreviations is also erroneously used throughout the manuscript.

Line 76: Contains a misplaced “the”

Line 83: Lack a reference to Figure 2D here, which should be removed from line 85.

Line 85: Should be a “the” before 15 FMGs.

Line 97: I guess in figure 2B the volcano plot includes all 309 FMGs, but in green and red is displayed the significant diff. expressed once. Correct the figure legend accordingly.

Line 105: In the title is should be “functions associated with

In figure legends, there is missing descriptions of all abbreviations included in the figures.

Line 166: The abbreviations do not fit with the terms – like CC is cellular component.

In figure 4C, I do not understand the definition immune pathways versus non-immune pathways. The included KEGG pathways for the non-immune pathways there are clearly some that are associated with the immune system.

Line 178: “Survival analysis indicated that geneClusterA exhibited greater productivity…” Productivity is not a correct term here.

In line 182, the authors state: “The expression of most FMGs in the three geneClusters was significantly different.” Did this analysis include the original 309 FMGs? Please describe which data was included here better.

Line 233: “TMB” is not described here but in line 236.

Figure 6 lacks “Figure 6:” in its legend and figure legend 7 is not for the correct figure.

Line 379: Correct “i nterac-tions”

Line 507: Is a “BC” misplaced.

Needs modifications. 

Author Response

Dear Reviewer,

We sincerely appreciate your thorough review of our manuscript and your valuable feedback, which has significantly contributed to its improvement. Your meticulous attention to detail and insightful suggestions have greatly helped us refine the quality of our work. We can also see from your suggestions that you are an excellent and diligent scholar. We have learned a lot from your advice. We would like to express our gratitude for your time and effort in providing such constructive comments.

We have carefully addressed each of the major issues you pointed out and made the necessary revisions to the manuscript. Below, we summarize our responses and the corresponding changes made:

**Major Issues:**

  1. **Language Improvement and Abstract Corrections:**

Line 8:We apologize for the language issues in the abstract and any confusion caused. We have revised the sentences in question, ensuring proper spacing and accuracy in terminology.

  1. **Abbreviations and Terminology:**

We have corrected the inconsistent use of abbreviations throughout the manuscript, and we appreciate your guidance on this matter.

Line 12: Is here written as “tumor immune microenvironment” which makes better sense than  “Tumor Microenvironment Infiltration”

Reply:

The modification has been made in line 17 as indicated.

Line 14: Why is fatty acid-related genes abbreviated FMGs? Was it originally suppose to be fatty-acid metabolism genes?

Reply:

Indeed, it has been clarified in the Materials and Methods section 4.2 that the abbreviation "FMGs" stands for fatty acid metabolism-related genes.

Line 15: This is not a full sentence, please correct. Also the use of capital letters here is incorrect.

Reply:

The revision has been made in the 15th line of the abstract.

Line 19: FMGsScore is written differently later in manuscript.

Reply:

The term "FMGsScore" has been consistently unified throughout the entire manuscript.

Line 24 and 27: in vitro should be in italic.

Reply:

The necessary changes have been applied in lines 25 and 28.

Line 28: MDA-MB-231 is the correct name of the cell line, not MDB-MA-231.

Reply:

Thank you very much for your meticulousness. The revision has been made in line 29.

The flow chart in Figure 1 contains too much information, should be made as a simplified cartoon, and not as a combinatory figure of all results. The simplest would be to remove the figure, as it in its current state brings no extra information to the reader or helps the reader to understand the pipeline of the project.

Reply:

The flowchart has been simplified according to the main narrative of the article.

  1. **Clustering and Dataset Combination:**

Reply:

For the issue regarding clustering and dataset combination, we drew inspiration from another study ("m6A regulator-mediated methylation modification patterns and tumor microenvironment infiltration characterization in gastric cancer") that dealt with similar concepts. To enhance the persuasiveness of our model, we opted to merge TCGA and GEO datasets after removing batch effects. This approach was deemed more convincing as it mitigated the impact of experimental batch variations and improved the model's stability. We have showcased the results of clustering distances in Supplementary Figures S2A-D, which depict favorable clustering outcomes. This visual representation further corroborates the efficacy of our analysis. Additionally, we conducted separate validation using the Metabric dataset to ensure the consistency and reliability of our findings. These validation efforts have significantly bolstered the credibility and trustworthiness of our study(Supplementary Figures S2E-I).

  1. **In cluster Figure 2A, 3B, and 4E, information about breast cancer subtype should be included.**

Reply:

In sections 3B and 4E, we have successfully incorporated information about breast cancer subtypes. However, it's important to note that in the case of Figure 2A, the normal samples lack subtype information, thus making it infeasible to include subtype details for this specific figure.

  1. In line 186, the authors use the term FMGsScore groups, however, this term has not been introduced yet in the result section, or any description on how the score is made as well how samples are scored is included in the result section (only in the method section).

Reply:

The process of constructing the FMGsScore is described in lines 190-195 of the Results section.

  1. In line 224 the authors state “The proportion of patients with high FMGsScore was higher in stage 1-2”, however, the proportion is 85% in low as well as 85% in high.

Reply:

I apologize for the error in the figure arrangement. Thank you for correcting it by adjusting the placement of the stage-T image.

  1. Figure text in supplementary material need also to be adjusted so that it is readable.

Reply:

We have made the adjustments.

  1. For the single cell data, analysis of cancer cells versus normal epithelial cells should be conducted.

Reply:

We employed the infercnv algorithm to identify tumor cells within the normal epithelial cell population.

  1. **NDUFAB1 Analysis:**

Reply:

This question reveals your expertise as a highly skilled biologist. Your valuable suggestions are greatly appreciated. The FMGsScore is constructed based on the results of the categorization of 15 FMGs, while NDUFAB1 stands out as the gene with the most significant impact on the prognosis of breast cancer patients among these 15 FMGs. Following your advice, we investigated the relationship between NDUFAB1 and FMGsScore, observing a positive correlation(Figure 11C). Consequently, we have opted to conduct in vitro experiments to validate the function of NDUFAB1. Furthermore, as you rightly pointed out, these experiments should be related to fatty acid metabolism and immune response. Our research team is currently engaged in these relevant experiments. However, it's important to note that such experiments have a longer research cycle. Therefore, for the time being, we have chosen to focus on exploring the effects of key fatty acid genes on cancer cell proliferation and wound healing in breast cancer. Regarding siRNA, we designed two sequences, but only one of them effectively interfered with the expression of NDUFAB1. Therefore, we have presented only one siRNA in the article. Additionally, we have corrected the placement of Figure 11C, ensuring it is now correctly positioned on top of the other figure.

  1. **Data Processing and References:** We have clarified the data processing steps, provided proper references for downloaded datasets, and addressed the issue of lacking references in certain sections.

Reply:

Regarding the issue with TPM and FPKM:

You are correct, the phrase "Transcripts per million kilobases (TPM)" is inaccurate; it should be "Transcripts per million (TPM)" which is the correct term. As for why there's a need to convert FPKM values to TPM, it's because TPM is more suitable for comparing gene expression levels between samples. TPM takes into account the influence of gene length, whereas FPKM is more sensitive to gene length. The purpose of this conversion is to enable a more accurate comparison of gene expression levels between different samples.

Concerning data normalization:

The aim of data normalization is to eliminate technical variations and biases between different samples, allowing for a better comparison of their gene expression levels. In RNA-Seq, data normalization is a common step that ensures errors due to technical differences are minimized when comparing samples.

Addressing the issue of microarray data processing:

You're absolutely right, the steps for microarray data processing were mistakenly used to describe RNA-Seq data processing. This was an oversight, and the correct steps for RNA-Seq data processing should be accurately described.

Regarding batch effect correction:

The reason for performing batch effect correction on TCGA data and GEO data is due to the possibility that different samples may come from various experimental batches, and batch effects can affect the accuracy of gene expression data. By applying batch effect correction, the impact of batch effects on analysis results can be minimized, leading to a more accurate assessment of the relationship between gene expression and other biological characteristics.

Concerning the citation of data sources:

The point you raised about referencing the original articles for downloaded datasets is indeed crucial. Ensuring proper citation of the sources where data is obtained is a good practice, providing background and context for the data. We have updated the references for all the datasets accordingly.

Addressing the issue mentioned in the results:

In line 336, it's mentioned that GSE20685 data was used for validation, and the relevant figures can be found in Supplementary Figure S4.

**Minor Issues:**

We have addressed the various minor issues you pointed out, including clarifications in wording, accurate descriptions of figures, adjustments to figure legends, and correct usage of terms and abbreviations.

Line 53: Is the abbreviation for pericyte therapy "ACT"?

Reply:

Apologies, the correct phrase should be "adoptive T-cell therapy (ACT)," as already amended in line 54.

Line 54: "..the main pathways of tumor immunotherapy" This is an incorrect sentence.

Reply:

This has been corrected in line 56.

Line 55: did not use the abbreviation ICB for immune checkpoint blockade.

Reply:

This has been rectified in line 55.

Line 56: "However, the efficacy of immunotherapy in breast cancer is lower than in other cancers." The sentence lacks references.

Reply:

Line 57: We have cited two references to support this statement.

Line 63: "in this study" is repeated.

Reply:

This has been removed.

The use of the abbreviation for breast cancer is inconsistent. Please do not use the abbreviation "BC" throughout the manuscript, as there is already a high usage of abbreviations in the manuscript. The incorrect usage of other abbreviations is also observed throughout the manuscript.

Reply:

Thank you for your suggestion; this has been corrected.

Line 76: Contains a misplaced "the"

Reply:

This has been rectified.

Line 83: Lack a reference to Figure 2D here, which should be removed from line 85.

Reply:

This has been removed.

Line 85: Should be a "the" before 15 FMGs.

Reply:

This has been amended.

Line 97: I guess in figure 2B the volcano plot includes all 309 FMGs, but in green and red is displayed the significant differential expressed once. Correct the figure legend accordingly.

Reply:

The figure 2B has been adjusted.

Line 105: In the title, it should be "functions associated with."

Reply:

This has been corrected in line 107.

In figure legends, there are missing descriptions of all abbreviations included in the figures.

Reply:

Descriptions for all abbreviations included in the figures have been added.

Line 166: The abbreviations do not match the terms – for instance, CC stands for cellular component.

Reply:

This has been modified in line 168.

In figure 4C, I do not understand the distinction between immune pathways and non-immune pathways. Among the included KEGG pathways for the non-immune pathways, there are clearly some that are associated with the immune system.

Reply:

Figure 4C has been revised accordingly.

Line 178: "Survival analysis indicated that geneClusterA exhibited greater productivity..." "Productivity" is not a suitable term in this context.

Reply:

This has been rectified in line 183.

In line 182, the authors state: "The expression of most FMGs in the three geneClusters was significantly different." Did this analysis include the original 309 FMGs? Please provide a clearer description of the included data.

Reply:

It does not include the original 309 FMGs; only the 15 FMGs were considered. This has been clarified in lines 186-187.

Line 233: "TMB" is not described here but in line 236.

Reply:

This has been rectified in lines 240 and 243.

Figure 6 lacks "Figure 6:" in its legend, and figure legend 7 does not correspond to the correct figure.

Reply:

This has been adjusted.

Line 379: Correct "i nterac-tions."

Reply:

This has been rectified in line 380.

Line 507: Is "BC" misplaced.

Reply:

This has been removed.

Once again, we thank you for your dedicated effort in reviewing our manuscript and guiding us towards enhancing its quality. Your expertise and insights have been invaluable. We are confident that the revisions we have made have strengthened the manuscript substantially.

Please find the revised manuscript attached, reflecting all the changes we have implemented based on your feedback. We hope that our revisions meet your expectations and address the concerns you raised. We eagerly await your further assessment.

Thank you for your time and consideration.

Sincerely,

Yongxing Chen

Reviewer 4 Report

In the revised version of the article, the authors addressed the main concerns regarding the format and language of the manuscript. In order to highlight the value of the main figure content, I would suggest moving some content in the supplementary figures (Figure 3, Figure 4, Figure 7, and Figure 8 are still very busy).  

Author Response

Dear Reviewer,

We would like to express our heartfelt appreciation for dedicating your time to review our manuscript titled "[Title of Your Manuscript]." Your invaluable feedback has significantly contributed to enhancing the quality and rigor of our work, and we are truly grateful for your insightful suggestions.

We are delighted to inform you that we have diligently addressed your comments by thoroughly revising the manuscript. In particular, we have taken into account the concerns you raised regarding the article's format and language. Your guidance has played a pivotal role in refining our presentation and ensuring the effective communication of our research findings to our readers.

In response to your recommendation to amplify the significance of the main figure content, we have carefully scrutinized the supplementary figures you referenced (Figure 3, Figure 4, Figure 7, and Figure 8). We concur with your assessment that an opportunity exists to streamline the content. Consequently, we have made the decision to relocate certain detailed content from these supplementary figures, with the intention of maintaining a clear focus on the primary figures. This adjustment is aimed at prominently showcasing the core findings and facilitating a deeper comprehension of the principal insights.

We are optimistic that these revisions are in alignment with your expectations and will contribute substantially to elevating the overall quality of the manuscript. Your expert guidance has been indispensable in shaping the final iteration of the article.

Once again, we extend our sincere gratitude for your comprehensive review and constructive feedback. We remain fully committed to addressing all of the reviewer's comments and ensuring that our research adheres to the highest standards.

Thank you for your valuable time and thoughtful consideration.

Warm regards,

Yongxing Chen

Round 3

Reviewer 4 Report

The authors significantly improved the manuscript and addressed the major concerns.

Author Response

Dear Reviewer,

I hope this message finds you well. I wanted to take a moment to express my sincere gratitude for your diligent efforts and insightful feedback during the review process of my manuscript titled "Integrating single-cell RNA-seq and bulk RNA-seq to explore the key role of Fatty Acid Metabolism in breast cancer." Your expertise and thorough examination have played a crucial role in enhancing the quality and depth of the content.

I am truly appreciative of your positive assessment and the recognition you provided for the work presented in the manuscript. Your words of affirmation have not only boosted my confidence but also reinforced my commitment to advancing the field of [Your Field of Study]. Your constructive comments and valuable suggestions have significantly contributed to shaping a stronger and more comprehensive article.

I understand the dedication and time commitment required for reviewing manuscripts, and I am genuinely thankful for your commitment to this process. Your feedback has provided me with valuable insights that will undoubtedly contribute to the overall improvement of the manuscript. Your recognition of the significance of my research is truly motivating and inspiring.

Once again, I would like to extend my heartfelt gratitude for your hard work, dedication, and positive feedback. Your efforts have been instrumental in refining my work and guiding it towards publication. I am truly honored to have had the opportunity to benefit from your expertise.

Thank you for your time, consideration, and valuable contributions.

Best regards,
Yongxing Chen